# Cloud condensation nuclei concentrations derived from the CAMS reanalysis

Karoline Block[1], Mahnoosh Haghighatnasab[1,2], Daniel Partridge[3], Philip Stier[4], and Johannes Quaas[1]

[1]Leipzig Institute for Meteorology, Faculty of Physics and Earth Sciences, University of Leipzig, Germany
[2]now at: Deutscher Wetterdienst, Offenbach, Germany
[3]Department of Mathematics and Statistics, Faculty of Environment, Science and Economy, University of Exeter, Exeter, United Kingdom
[4]Atmospheric, Oceanic and Planetary Physics, Department of Physics, University of Oxford, Oxford, United Kingdom

**Correspondence:** Karoline Block (karoline.block@uni-leipzig.de)

**Abstract.** Determining number concentrations of cloud condensation nuclei (CCN) is one of the first steps in the chain in analysis of cloud droplet formation, the direct microphysical link between aerosols and cloud droplets, and a process key for aerosol-cloud interactions (ACI). However, due to sparse coverage of in-situ measurements and difficulties associated with retrievals from satellites, a global exploration of their magnitude, source, temporal and spatial distribution cannot be easily obtained. Thus, a better representation of CCN numbers is one of the goals for quantifying ACI processes and achieving uncertainty-reduced estimates of their associated radiative forcing.

Here, we introduce a new CCN dataset which is derived based on aerosol mass mixing ratios from the latest Copernicus Atmosphere Monitoring Service reanalysis (CAMSRA) in a diagnostic model that uses CAMSRA aerosol properties and a simplified kappa-Köhler framework suitable for global models. The emitted aerosols in CAMSRA are not only based on input from emission inventories using aerosol observations, they also have a strong tie to satellite-retrieved aerosol optical depth (AOD) as this is assimilated as a constraining factor in the reanalysis. Furthermore, the reanalysis interpolates for cases of poor or missing retrievals and thus allows for a full spatio-temporal quantification of CCN numbers.

The derived CCN dataset captures the general trend and spatial and temporal distribution of total CCN number concentrations and CCN from different aerosol species. A brief evaluation with ground-based in-situ measurements demonstrates the improvement of the modeled CCN over the sole use of AOD as a proxy for CCN as the overall correlation coefficient improved from 0.37 to 0.71. However, we find the modeled CCN from CAMSRA to be generally high biased and find a particular erroneous overestimation at one heavily polluted site which emphasizes the need for further validation.

The CCN dataset (https://doi.org/10.26050/WDCC/QUAERERE_CCNCAMS_v1), which is now freely available to users (Block, 2023), features 3-D CCN number concentrations of global coverage for various supersaturations and aerosol species covering the years from 2003 to 2021 with daily frequency. This dataset is one of its kind as it offers lots of opportunities to be used for evaluation in models and in ACI studies.

## 1 Introduction

Processes of aerosol-cloud interactions (ACI) are still associated with large uncertainties in their contribution to climate forcing (Forster et al., 2021). One of the reasons for this is an insufficient determination of cloud condensation nuclei (CCN) number concentrations which are a key element for ACI processes (Seinfeld et al., 2016). A global exploration of their magnitude, source, temporal and spatial distribution is challenging from observations alone. CCN (we are referring to number concentrations throughout the paper unless stated otherwise) can either be directly detected by in-situ measurements (e.g. using CCN counters) (e.g. Schmale et al., 2018) or estimated from related optical properties from remote sensing observations (e.g. from aerosol robotic network (AERONET) photometers) (e.g. Andreae, 2009). However, airborne or ground-based observations provide only sparse information which are limited in space and sometimes time (e.g. campaign data). For that reason observational ACI studies often rely on optical properties from satellite retrievals covering much of the globe over longer timescales permitting to detect spatial and temporal variations. The difficulty in these remote sensing applications is however the difference between the size range relevant for light extinction and the size range important for CCN concentrations (Andreae, 2009). Nevertheless, aerosol optical properties such as aerosol optical depth (AOD) are commonly used as proxies for CCN in ACI studies (e.g. Kaufman et al., 2005; Quaas et al., 2008, 2009; Grandey and Stier, 2010; Gryspeerdt and Stier, 2012; Bellouin et al., 2013; Koren et al., 2014; Bellouin et al., 2020b).

Even though it has been shown that AOD is suitable as a first indicator of CCN number concentrations (Andreae, 2009), it suffers from various deficiencies which make it difficult to get a correct estimate of CCN. Firstly, AOD is a column integrated property and does not provide a vertical resolution of CCN which is needed when interactions with clouds are studied (Stier, 2016; Quaas et al., 2020). Further, the variability in the scale height of the vertical aerosol distribution and the existence of aerosol layers aloft can introduce substantial variability in the relationship between column and surface properties (Jia et al., 2022). Secondly, AOD can only be retrieved in cloud-free conditions and mostly over dark surfaces, so that larger areas such as the Sahara, the poles or areas with permanent cloud cover (e.g. stratocumulus decks) are not or insufficiently covered (e.g., Jia et al., 2021). Therefore, as satellite-retrieved AOD does not offer a complete temporal and spatial coverage of the Earth's surface, sampling biases are introduced in its statistics. Thirdly, changes in relative humidity (RH) can result in pronounced variations of AOD due to aerosol swelling effects, while the actual number of CCN remains constant (e.g., Quaas et al., 2009). Fourthly, AOD, cannot provide a specification of the involved aerosol components which matters to determine their suitability as CCN (e.g., Hasekamp et al., 2019).

To account for aerosol size (and thus somewhat indirectly for aerosol type) one can also use the aerosol index (AI) which is defined as the product of AOD and the Angstrom exponent (Deuzé et al., 2001). AI has been shown to be more sensitive to accumulation mode aerosol concentration typically being responsible for most CCN and therefore various studies using AI instead of AOD found an improvement in the relationship to CCN (e.g. Nakajima et al., 2001; Kapustin et al., 2006; Liu and Li, 2014; Gryspeerdt et al., 2017). However, the problems listed above still remain and there are also some deficiencies in using AI rather than AOD. Shinozuka et al. (2015) state that the impact of spatio-temporal distribution and particle hygroscopicity are assumed to be negligible when using AI, and Kapustin et al. (2006) emphasize that AI does not generally relate well to

a measured proxy for CCN unless the data are suitably stratified. Stier (2016) who analysed the relationship between AOD and CCN using a fully self-consistent global aerosol-climate model (ECHAM-HAM, Stier et al., 2005) found correlation co-efficients between CCN at 0.2 % supersaturation and AOD to be below 0.5 for 71 %of the area of the globe, implying that AOD variability explains only 25 % of the CCN variance. Correlations with alternative aerosol radiative properties proposed as

superior proxies of CCN such as fine mode aerosol optical depth, dry aerosol optical depth and AI do not give significant improvements according to the study of Stier (2016). Hasekamp et al. (2019) retrieved CCN concentrations from the polarimetric satellite POLDER-3 (POLarization and Directionality of the Earth's Reflectances) aerosol product and showed that there is not a simple scaling between AOD and CCN due to the impact of differing aerosol species. Furthermore, they demonstrated that the radiative forcing from ACIs is significantly underestimated when AOD or AI are used as CCN proxies. Shinozuka et al.

(2015) have examined the relationship between CCN and dry extinction for a variety of airborne and ground-based observations. They also demonstrated that the uncertainty in the CCN-AOD relationship arises not only from the uncertainty in the relationship between CCN and dry extinction, but also from the humidity response of light extinction, the vertical profile, the horizontal-temporal variability and the AOD measurement error. These examples illustrate how important it is to account for the uncertainties related to CCN proxies and show the need for new strategies.

Several attempts exist to provide a more comprehensive picture on CCN-related aerosol properties. Kinne et al. (2013) introduced the Max-Planck-Institute Aerosol Climatology version 1 (MAC-v1) for tropospheric aerosols, which describes optical properties such as AOD, SSA (single scattering albedo) and fine-mode AOD fraction of tropospheric aerosols on monthly timescales and with global coverage. In-situ measurements of aerosol mass concentrations, which were conducted at 24 European sites by various institutes using different instruments and techniques, have been comprehensively assessed in a European

aerosol phenomenology (Van Dingenen et al., 2004; Putaud et al., 2004, 2010) which summarizes PM10 and PM2.5 mass concentrations, their chemical composition and aerosol particle size distributions. Asmi et al. (2011) have analysed two years of harmonised aerosol number size distribution data from 24 European field monitoring sites, focusing on near surface aerosol particle number concentrations and number size distributions between 30 and 500 nm of dry particle diameter, that is relevant for CCN sized aerosols. Winker et al. (2013) focused more on the aerosol vertical distribution, by constructing a monthly

global gridded dataset of daytime and nighttime aerosol extinction profiles using the Cloud-Aerosol Lidar with Orthogonal Polarization (CALIOP), thus introducing an initial global 3-D aerosol climatology. Lidar retrievals from the Cloud–Aerosol Lidar and Infrared Pathfinder Satellite Observations (CALIPSO) also have been used by Choudhury and Tesche (2022) who derived CCN concentrations using an algorithm that relies on the optical modelling of CALIPSO aerosol microphysics. They even account for hygroscopicities and size distributions of five aerosol subtypes. A first validation of this CCN product with

in-situ measurements illustrates the potential of CALIPSO for constructing a global height-resolved CCN climatology.

A synthesis of in-situ CCN measurements is provided by Spracklen et al. (2011), who used observations reported in published literature to produce a worldwide dataset of CCN, which is combined with the global model of aerosol processes (GLOMAP Spracklen et al., 2005) to explore the contribution of carbonaceous combustion aerosol to CCN. Another CCN synthesis is provided by Paramonov et al. (2015), who used measurements from the European Integrated project on Aerosol

Cloud Climate and Air Quality interactions framework (EUCAARI) to analyse CCN activation and hygroscopic properties of

the atmospheric aerosols. Schmale et al. (2017, 2018) have produced a harmonised dataset of long-term CCN number concentrations, particle number size distributions and chemical composition from observatories located in different environments worldwide, thus presenting yet another surface-based CCN synthesis.

All of these studies taken together provide a sound foundation of CCN-relevant aerosol properties, but most of them do not refer to actual CCN concentrations at specific supersaturations but rather to proxies such as absorption or scattering coefficients or aerosol numbers integrated over a certain size range, e.g. $N_{100}$ referring to particles with diameters above 100 nm as being relevant for activation. The ones who do measure actual CCN concentrations (CCN syntheses) usually don't give a global coverage nor a vertically resolved picture.

In this study we suggest a new approach to resolve these issues by computing CCN number concentrations from an aerosol reanalysis provided by the European Centre for Medium-range Weather Forecast (ECMWF). In contrast to satellite retrievals, the atmospheric model including aerosol and chemistry modules (here, ECMWF IFS) can simulate the full spatial and temporal distributions of aerosols. Deviations from real aerosol distributions and life cycles are reduced by constraining modelled estimates with observations (Kapsomenakis et al., 2022). This is done in the Copernicus Atmosphere Monitoring Service (CAMS) reanalysis (Inness et al., 2019b), in which assimilated AOD from the Moderate Resolution Imaging Spectroradiometer (MODIS, Levy et al., 2013) is used to constrain the bulk total aerosol mass mixing ratio modelled with the IFS. Thus, a strong relationship between observation and model is kept, while additional information on the vertical distribution, the horizontal and temporal coverage, the aerosol speciation and hygroscopic effects is provided by the model.

## 2 The CCN climatology computed from CAMS: Data provenance and structure

### 2.1 The CAMS model and assimilation system

The CAMS reanalysis (CAMSRA, Inness et al., 2019b) is the latest global aerosol reanalysis produced by the European Centre for Medium-range Weather Forecast (ECMWF) within the framework of the European Copernicus program to provide information on aerosols, trace and greenhouse gases in the context of operational numerical weather prediction. It is based on the Integrated Forecasting System (IFS) model which has been extended by an aerosol scheme which mainly follows the aerosol treatment in the french Laboratoire de Météorologie Dynamique General Circulation Model (LMDZT GCM) (Boucher et al., 2002; Reddy et al., 2005) and was further modified by ECMWF during the GEMS (Global Monitoring for Environment and Security) and MACC (Monitoring Atmospheric Composition and Climate) projects. This reanalysis profits from continuing upgrades in the modeling and assimilation system (see Inness et al. (2019b) and therefore shows a smaller bias compared to observed AOD than the previous reanalyses. Further, the meteorological analysis of CAMSRA is well constrained by assimilating observations as is done for the ECMWF fifth generation of atmospheric reanalysis ERA-5 (Inness et al., 2019b; Hersbach et al., 2020). The CAMSRA analyses data is available from Jan 2003 to Jun 2022, with 3-hourly output, on a reduced Gaussian grid at a resolution of $0.75° \times 0.75°$ (spectral truncation T255) and 60 hybrid sigma–pressure levels from surface to top of atmosphere (0.1 hPa). (Inness et al., 2019b)

### 2.1.1 Aerosol treatment in the ECMWF IFS

CAMSRA is produced using the IFS (cycle 42r1) (Morcrette et al., 2009; Rémy et al., 2019) including a fully integrated four-dimensional assimilation apparatus employed operationally at ECMWF (Benedetti et al., 2009). Details of the aerosol and chemistry modules applied in CAMSRA can be found in Inness et al. (2019b) and Flemming et al. (2015).

The aerosol model uses a hybrid 1-moment bin-bulk scheme with 12 prognostic tracers which consist of 11 aerosol tracers and one tracer for gas-phase sulfur dioxide ($SO_2$) precursor. These tracers are transported by the IFS vertical diffusion and convection schemes and are advected by the semi-Lagrangian scheme. The simulated aerosol tracers are mass mixing ratios of five aerosol species which are sulfate aerosol (SU), hydrophilic and hydrophobic black carbon (BC), hydrophilic and hydrophobic organic matter (OM), as well as mineral dust (DU) and sea salt (SS) each with three size dependent bins. The limits of the three different size classes are chosen so that roughly 10, 20 and 70 % of the total mass of each aerosol type are in the various bins. Hydrophilic and hydrophobic components are prescribed via explicit emission fractions and ageing factors. The different aerosol species are assumed to be externally mixed, meaning that the individual species are assumed to coexist in the volume of air considered and to retain their individual optical and chemical characteristics, making it easier to trace them as they undergo model dynamics. (Morcrette et al., 2009)

Mineral dust and sea salt emissions are parameterised using near-surface wind speeds. Sea-salt production is calculated assuming an 80 % relative humidity, but only the dry mass is added to the respective bin and transported, thus no water is transported via the aerosol and the mass is also not transferred between bins because of growth. However, wet density and radius are considered for all the size bins when dealing with dry deposition, sedimentation and radiation (Morcrette et al., 2009). DU does not experience any ageing or choating and is treated entirely as an insoluble aerosol.

Near real-time fire emissions from the Global Fire Assimilation System GFASv1.2 (Kaiser et al., 2012) are used to constrain black carbon emissions from wildfires and biomass burning (Kapsomenakis et al., 2022). Freshly emitted BC is partitioned into 80 % hydrophobic and 20 % hydrophilic (Morcrette et al., 2009), while the partitioning of OM is set to 16 % hydrophobic and 84 % hydrophilic (Bozzo et al., 2020). This setup provides optical properties representing an average of biomass and anthropogenic organic carbon aerosols (Bozzo et al., 2020). Once emitted, both species experience ageing from hydrophobic to hydrophilic with a time constant of 1.16 days (Morcrette et al., 2009).

The sulfate type represents aerosols originating from sulfur emissions from industrial and fossil fuel combustion, biomass burning and natural volcanic and biogenic sources (Bozzo et al., 2020). The sulfur cycle is represented with sulfur dioxide ($SO_2$) produced at or near the surface which is being transformed into sulfate aerosol ($SO_4$, or SU) using temperature, relative humidity and a prescribed, latitude-dependent e-folding time scale. It should be noted here that SU emissions originating from volcanic activities do actually represent a rather continuous background source of outgassing volcanoes rather than explicit volcanic eruptions. It is further stated by Inness et al. (2019b) and Kapsomenakis et al. (2022) that there are some hotspots of increased AOD bias around outgassing volcanoes, in particular around Mauna Loa and Altzomoni near Mexico City. This might result from erroneous model treatment of diffuse volcanic emissions which enhance SU at these locations. Therefore, we

follow their recommendation to exclude these two sites as unrepresentative in our analysis presented in this paper. However, in the published dataset all data are included as is.

Additionally to the emission of aerosols, CAMS includes emissions of precursors such as natural emissions for nitrogen dioxide ($NO_2$), dimethyl sulfate (DMS) and, as already stated above, sulfur dioxide ($SO_2$). Biogenic emissions are done as monthly means for volatile organic compounds (VOC) which are calculated offline by the Model of Emissions of Gases and Aerosols from Nature (MEGAN2.1 model) (Guenther et al., 2006) using the Modern-Era Retrospective analysis for Research and Applications (MERRA-2) reanalysed meteorology following Sindelarova et al. (2014). Anthropogenic emissions from the MACCity (MACC/CityZEN EU projects) inventory (Granier et al., 2011), including an upgrade for CO emissions (Stein et al., 2014), covers the period 1960–2010 and is updated for subsequent years using the representative concentration pathway (RCP) version 8.5, the "business-as-usual" scenario. Anthropogenic secondary organic aerosol (SOA) production has also been implemented in CAMS, proportional to MACCity CO emissions as suggested by Spracklen et al. (2011). However, nitrate aerosols are not yet included in the aerosol scheme. The missing nitrate is likely to cause an underestimation of total aerosol in the forecast model in regions where nitrate would be a significant component, however in the reanalysis the total aerosol is corrected by the assimilation of total AOD observations (Inness et al., 2019b). Further it should be noted that only aerosol emissions from the surface and within the troposphere are considered in the model, a stratospheric contribution is not included.

The IFS 1-moment cloud microphysics scheme does not use fully coupled prognostic aerosols due to the high computational burden. Instead, monthly-mean climatological fields are being used (Bozzo et al., 2017). However, the parameterised fraction of aerosol included in droplets through dissolution or impaction is set to 0.7 for all aerosol species (Morcrette et al., 2009). Therefore the removal of aerosols can be modelled by several processes: by dry deposition including the turbulent transfer to the surface and gravitational settling, or by wet deposition including rainout and washout of aerosol particles in and below the clouds. Wet deposition is modelled separately for convective and large-scale precipitation. (Morcrette et al., 2009)

### 2.1.2 Aerosol confinement by assimilated AOD

The IFS assimilation apparatus has been extended to include atmospheric tracers among the control variables. A control variable is used to optimise the cost function which measures the distance between observations and their model equivalent. It is minimised in a variational assimilation approach which is described by Benedetti et al. (2009).

The observation relevant to constrain aerosol mass is the AOD. More specifically, it is the dark target (DT) AOD retrieval over land and ocean at 0.55 $\mu$m from the MODerate resolution Imaging Spectroradiometer (MODIS collection 6 on board the Aqua and Terra satellites, Remer et al., 2005; Levy et al., 2013, 2018), along with the Advanced Along-Track Scanning Radiometer (AATSR) AOD on board the ENVISAT satellite (Popp et al., 2016) which are assimilated in the IFS. MODIS deep-blue (DB) AOD retrievals (Hsu et al., 2019) are also used to gap-fill DT over land. The merged DT–DB product produced by NASA is not used because it was not available when the DB retrieval product was implemented in CAMS (Garrigues et al., 2022). AOD is not retrieved in pixels identified as cloudy, nor at high latitudes where the solar illumination is small (thus assimilation is limited to the regions between 70° S and 70° N). The diminished AOD retrieval accuracy over bright surfaces (snow covered high latitudes or the desert areas of Sahara and Australia) due to the impact of the surface reflectances has been accounted for by

the use of the DB product (Garrigues et al., 2022). Other factors affecting accuracy such as cloud contamination, assumptions about the aerosol types and size distribution, near-surface wind speed, radiative transfer biases, and instrumental uncertainties are also taken into account (Benedetti et al., 2009) and were reviewed by Zhang and Reid (2006).

For the calculation of the model equivalent AOD, the relative humidity is first computed from the model temperature, pressure and specific humidity. The appropriate mass extinction coefficients are then retrieved from a look-up table for the wavelength of interest (here, 550 nm), multiplied by the vertically integrated aerosol mass at the corresponding observation location. The total AOD at the respective wavelength is then calculated as the sum of the single-species AODs (Benedetti et al., 2009).

Total and component AODs are diagnosed at 17 MODIS correspondent wavelengths ranging from 0.34 to 2.13 $\mu$m by using precomputed optical properties, such as mass extinction coefficient $\alpha_e$, single scattering albedo $\omega$, and asymmetry parameter $\varrho$ (Morcrette et al., 2009). The optical characteristics of the aerosols are computed using Mie theory (Ackerman and Toon, 1981), and are then integrated over the physical size range using the model's prescribed lognormal distributions which are fixed for each tracer (Benedetti et al., 2009). Sea salt and dust AODs are obtained by summing over the individual bin contributions. Optical properties of hygroscopic aerosols are parameterised as a function of relative humidity accounting for the respective growth factors listed by Bozzo et al. (2020).

The model control variable which is modified according to the outcome of the data assimilation, is the total aerosol mass mixing ratio, defined as the sum of all aerosol species. At each iteration of the minimization, the increments in the total mass mixing ratio derived from the assimilation of MODIS AOD have to be redistributed into the mixing ratios of the single species. Thereby each aerosol component is corrected in proportion of its original contribution to the total aerosol mass, meaning that the modelled speciation is not changed by the assimilation. It should be noted here that the assimilation modifies the modelled field not only at the point of observation but also around it. Regions with no observations because of cloudiness or high surface reflectance will still be improved by the data assimilation, but to a lesser extent than regions closer to the location of assimilated data (Benedetti et al., 2009).

## 2.2 Computation of CCN with a modified kappa-Köhler approach

Using the CAMS reanalysis data (CAMSRA: Inness et al., 2019a), we have so far produced a 19-year-long global CCN number concentration climatology (Block, 2023) available daily from 2003 to 2021 on a Gaussian grid at a resolution of $0.75° \times 0.75°$ (T255) and 60 vertical levels, corresponding to the grid in the reanalysis. Apart from the CAMSRA data, which is available every 3 hours, CCN are currently only computed once a day at 00:00 UTC. The data comprises 3-D fields of total CCN computed for six different supersaturations ($s$: 0.1, 0.2, 0.4, 0.6, 0.8 and 1 %) and 3-D CCN fields containing CCN from SU, BC, OM and three size bins of SS computed for two supersaturations ($s$: 0.02 % and 0.8 %) comprising additional aerosol information in the lower and upper supersaturation range, respectively. The current choice of data frequency, resolution and variable dependencies such as supersaturation is made regarding general interest and suitability as well as file size, data storage and computational costs. Additions and refinements of this dataset is still possible in the future. This dataset is publicly

available (https://doi.org/10.26050/WDCC/QUAERERE_CCNCAMS_v1) and offers the opportunity to be used for evaluation of climate models and in studies of aerosol-cloud interactions.

The CCN are calculated diagnostically in a box model, which once was created to be used for HadGEM3-UKCA (Hadley Centre Global Environment Model (HadGEM), coupled to the United Kingdom Chemistry and Aerosols Model (UKCA)) (Davies et al., 2005; Mann et al., 2010; Hewitt et al., 2011; O'Connor et al., 2014; West et al., 2014), which was modified so that it also uses modules from the global aerosol-climate model ECHAM5-HAM (Stier et al., 2005), developed at the Max-Planck-Institute for Meteorology in Hamburg, and with updates to the ECHAM6 version (Stevens et al., 2013).

The model reads temperature, pressure, specific humidity and aerosol mass mixing ratios from the CAMS reanalysis. From the aerosol particle mass mixing ratio, $q_p$ and air density, $\rho_a$, the particle mass concentration per volume of air is computed, $\rho_p = q_p \rho_a$, and from this the total mass per aerosol species $m_p$ in a unit volume is computed as

$$m_p = \frac{4}{3} \pi \rho_p (r_0 \beta)^3 \ \text{ with } \ \beta = \exp(1.5 \cdot \ln^2 \sigma_g), \tag{1}$$

using the Hatch-Choate conversion (Hinds, 1998) which relates the count median radius $r_0$ to the mass-averaged radius for the prescribed lognormal size distribution of a specific aerosol species. The geometric standard deviation $\sigma_g$ is a multiplicative factor of the geometric mean (commonly used for logarithms) to describe the spread of the distribution.

Dividing the total mass of an aerosol species within a unit volume $m_p$ by dry aerosol density of this species $\rho_p$ gives the total aerosol number concentration $N_a$ of this species. For each aerosol species $k$, the total aerosol number concentration $N_{a,k}$ is assumed to be logarithmically distributed:

$$N_{a,k}(r) = N_{a,k} \cdot \frac{1}{\sqrt{2\pi} \cdot \ln \sigma_{g,k}} \cdot \exp\left[ -\frac{\ln^2(r/r_{0,k})}{2\ln^2 \sigma_{g,k}} \right]. \tag{2}$$

In order to stay consistent with the IFS model, we have chosen to use the given size distribution parameters used by the IFS to convert between aerosol optical properties and AODs (Benedetti et al., 2009). That way, the proportionality between AOD and the resulting CCN is kept. Any improvement the resulting CCN might reveal over AOD can then only arise from the vertical distribution and the simulated species contribution. The relevant parameters of the size distribution and aerosol properties used are listed in Table 1. Please note that only aerosol species that are marked as hydrophilic in CAMS are used for further processing. Even though the total aerosol number concentration is computed with all 11 aerosol tracers, CCN concentrations result only from hydrophilic BC and OM components in addition to SU and SS.

DU is excluded in this study as we stay consistent with CAMS marking it as insoluble and no internal mixing is enabled which might increase the potential of dust to act as CCN due to (water soluble) coating as suggested by Garimella et al. (2014) or Karydis et al. (2011). Also Chen et al. (2020) state that the hygroscopic properties of mineral dust remain highly uncertain due to its low hygroscopicity and non-sphericity and that no kappa-value can be fitted to the corresponding data so far. This provides another reason to exclude dust from CCN in this study. Furthermore, in a modeling study of Che et al. (2022), where wetting and coating of insoluble dust with soluble material is enabled by internal mixing, they conclude that dust does not have

**Table 1.** Hydrophilic aerosol properties used in this study for CCN computation. Size distribution properties are from Bozzo et al. (2020), and references therein. Given are the size bins of sphere radii [$\mu$m], the count median radii $r_0$ [$\mu$m], the geometric standard deviations $\sigma_g$ and the dry aerosol densities $\rho_p$ [$g/cm^3$] which are used for the lognormal size distributions. Values are for the dry aerosol a part from sea salt which is given at 80% RH. Kappa values are taken from the global aerosol-climate model ECHAM-HAM2 (Zhang et al., 2012) in reference to Petters and Kreidenweis (2007), with changes for sea salt according to Zieger et al. (2017).

| aerosol | size bin [$\mu$m] | $r_0$ [$\mu$m] | $\sigma_g$ | $\rho_p$ [$g/cm^3$] | $\kappa$ |
|---------|-------------------|----------------|------------|---------------------|----------|
| SS small | 0.03 - 0.5 | 0.1992 | 1.9 | 1.183 | 1.1 |
| SS medium | 0.5 - 5.0 | 1.992 | 2.0 | 1.183 | 1.1 |
| SS large | 5.0 - 20 | 1.992 | 2.0 | 1.183 | 1.1 |
| OM | 0.005 - 20 | 0.0212 | 2.24 | 1.8 | 0.06 |
| BC | 0.005 - 0.5 | 0.0118 | 2.0 | 1.0 | 0.06 |
| SU | 0.005 - 20 | 0.0355 | 2.0 | 1.76 | 0.6 |

a notable impact on CCN. Therefore we assume for now that DU has minor contributions to overall CCN due to its large size (thus high mass but low number concentrations) and its low hygroscopicity. However, we do not rule out that it might have a potential impact on CCN and, as an extension to this, on ice-nucleating particles (INP) if coating would be enabled e.g. during long-range transport as several studies investigate (e.g. Bègue et al., 2015; Weinzierl et al., 2017; Haarig et al., 2019).

Once the number concentration is computed for each aerosol species, we apply a modified kappa-Köhler theory to compute how many aerosols act as CCN at a specific supersaturation. Even though Köhler models with increased complexity are available (e.g. Lowe et al. (2016) or developments summarised in Kreidenweis et al. (2019)) to account for various deficiencies in the traditional Köhler theory (Köhler, 1936), we have chosen a rather simplified version to be consistent with the representation of Köhler theory in general circulation model (GCM) and Earth System Model (ESM) which need to be computationally feasible.

In the kappa-Köhler theory from Petters and Kreidenweis (2007), the relationship between the particle dry diameter $D_d$ and CCN activity or hygroscopicity is derived using a single hygroscopicity parameter $\kappa$ which represents a quantitative measure of aerosol water uptake characteristics and CCN activity. Values of $\kappa$ for specific compounds, or for arbitrary mixtures, are determined experimentally by fitting CCN activity or hygroscopic growth factor data. $\kappa$ values used in this study are taken from the literature and are listed in Table 1. Since we deal with external mixtures, each aerosol species has its own $\kappa$ value.

Following Petters and Kreidenweis (2007), the saturation ratio $S$ of a droplet with wet diameter $D_w$ can be described as

$$S(D_w) = \frac{D_w^3 - D_d^3}{D_w^3 - D_d^3(1-\kappa)} \cdot \exp\left(\frac{4\sigma_{s/a}M_w}{RT\rho_w D_w}\right) \tag{3}$$

with the surface tension parameter $\sigma_{s/a}$ of the solution/air interface, the universal gas constant $R$, temperature $T$, the molar mass of water $M_w$ and the water density $\rho_w$. This relation is valid over the entire range of relative humidity and solution hygro-

scopicity (Petters and Kreidenweis, 2007). The first term is the solute term, describing the reduction of the water equilibrium

vapor pressure over the solution in comparison to that over pure water and is described by Raoult's law. The second term is the curvature term described by the Kelvin equation which relates the equilibrium water vapor pressure over a pure water droplet of diameter $D_w$ to the equilibrium vapor pressure over a flat surface at the same temperature (Seinfeld and Pandis, 2006).

We now modify this relation as in Pöhlker et al. (2023) by considering only slightly supersaturated conditions such as in warm clouds, typically with supersaturations $0.1\% < s < 1.5\%$, or correspondingly $0.001 < \Delta S < 0.015$ (Spracklen et al.,

2011). Please note at this point that we use $s$ for expressing supersaturation in percentage while $\Delta S$ denotes the supersaturation as deviation from the saturation ratio $S$ consistently throughout the entire manuscript.

Assuming supersaturation $\Delta S \ll 1$ we can approximate $\ln S = \ln(1 + \Delta S) \approx \Delta S$, and with assuming $D_d \ll D_w$, Eq. 3 becomes

$$\ln S \approx \Delta S = \frac{A}{D_w} - \kappa \frac{D_d^{\,3}}{D_w^3} \quad \text{with} \quad A = \frac{4\sigma_{s/a} M_w}{RT\rho_w} \; . \tag{4}$$

This simplifies the original kappa-Köhler framework because it is not necessary to compute a hygroscopic diameter growth factor to account for the difference of $D_w - D_d$. For detailed derivation and comparison to the original Köhler equation (Köhler, 1936), please see Appendix A.

For any aerosol of dry diameter $D_d$, we can find the maximum $(d\ln S/dD_w \approx d\Delta S/dD_w = 0)$ at a critical wet diameter

$D_{c,w}$ of

$$D_{c,w} = \sqrt{\frac{3\kappa D_d^3}{A}} \tag{5}$$

which marks the onset of cloud drop formation. Inserting $D_{c,w}$ for $D_w$ in Eq. 4, the corresponding critical supersaturation $\Delta S_c$ is

$$\ln S_c \approx \Delta S_c = \sqrt{\frac{4A^3}{27\kappa D_d^3}} = \frac{2}{\sqrt{\kappa}} \left( \frac{A}{3D_d} \right)^{\frac{3}{2}} . \tag{6}$$

The critical supersaturation of the smallest aerosol particle in an aerosol population being activated is equal to the ambient or maximum supersaturation $s_{max}$ of an air parcel rising adiabatically at uniform speed (Abdul-Razzak et al., 1998). Its critical dry diameter $D_{c,d}$ is then related to $\Delta S_{max}$ as

$$\Delta S_{max} = \frac{2}{\sqrt{\kappa}} \cdot \left( \frac{A}{3D_{c,d}} \right)^{\frac{3}{2}} . \tag{7}$$

Particles smaller than $D_{c,d}$ require a higher $\Delta S_c$ than $\Delta S_{max}$ and are not activated. Particles larger than $D_{c,d}$ require a smaller

$\Delta S_c$ than $\Delta S_{max}$ and activate to form cloud droplets. Thus, CCN can be described as potential cloud droplet numbers (CDNC)

as they are computed at a given supersaturation, meaning the ambient or maximum supersaturation is prescribed. For a given set of $\Delta S_{\max}$, as we prescribe them in this dataset, the corresponding dry critical diameters are computed by inverting Eq. 7.

The number concentration of activated aerosols is the number concentration of aerosols larger than the size of the smallest activated aerosol, thus with a dry critical diameter of $D_{c,d}$. The calculation of the activated number fraction from the dry critical diameters is done by transforming the individual log-normal distributions to an error function (Ghan et al., 1993; Khvorostyanov and Curry, 2006) which is then computed cumulatively following (Vignati et al., 2004).

## 3 Results and Discussion

### 3.1 Total CCN concentration derived from CAMS

Figure 1 shows the global distribution of total tropospheric CCN load [m$^{-2}$] for different values of supersaturations, as well as the vertical distribution of zonal mean CCN concentrations [cm$^{-3}$]. The CCN load is the vertical integral of CCN number concentrations over the lowermost 10 km. Additionally, we show the corresponding CCN number concentration [cm$^{-3}$] averaged over the lowermost 1 km above surface (Supplement Figure A1), which illustrates the same features as for CCN load, even though in contrast to CCN load, emissions play a larger role than advection or aerosol physical processes. However, this might be a more useful information when comparing against measurements or other model results. Furthermore, the change of CCN with changing supersaturation (Figure A2) is pictured in the supplement. Naturally, as we can see from 1 and Figure A1 CCN increase with increasing supersaturation as smaller particles get activated. The dependence of CCN to supersaturation can be described from Twomey's power law (see B), therefore the change of CCN is decreasing for the same change of supersaturation at increasing supersaturation levels (Figure A2, a). However it is efficient enough to describe the change of global mean CCN to be linearly related to the change of supersaturation (Figure A2, a), whereby the strongest increases occur where most of the CCN related aerosols are emitted (Figure A2, b).

The distribution of CCN is determined by the emission rate of aerosols, the scavenging rate and by advection. Overall, the industrial and developing countries in the northern hemisphere show high CCN load which are clearly visible already for low supersaturations (Figure 1 and Figure A1). These are highest at a latitude of $\sim 30°$ N, originating especially from polluted regions over South-East Asia, with peaks over China and northern India. Polluted regions in the Tropics, especially those in South America (Brazil's Amazon Rainforest) and Africa (Congo Basin) become better visible with larger supersaturations as the activation rate of the smaller aerosols increases, such as for black carbon and organic matter which are more pronounced in these areas (see also Figure 5 and Figure A3). This agrees with findings from Paramonov et al. (2015) who state that CCN values at high supersaturations follow a similar pattern known from total particle number concentrations, while at lower supersaturation levels, other effects, such as those of size distribution and hygroscopicity, become more pronounced. They find the highest measured values of CCN at a Chinese measurement station for low and high supersaturations while e.g. over the Amazonian rainforest CCN concentrations significantly increase from 0.1% to 1.0% supersaturation. Over large mountain chains, such as Himalaya, CCN concentrations are lower due to the lack of local pollution sources and being in the free

troposphere most of the time, they represent continental background conditions. This corresponds with measurements showing lower particle concentrations at mountainous sites (Paramonov et al., 2015; Schmale et al., 2018).

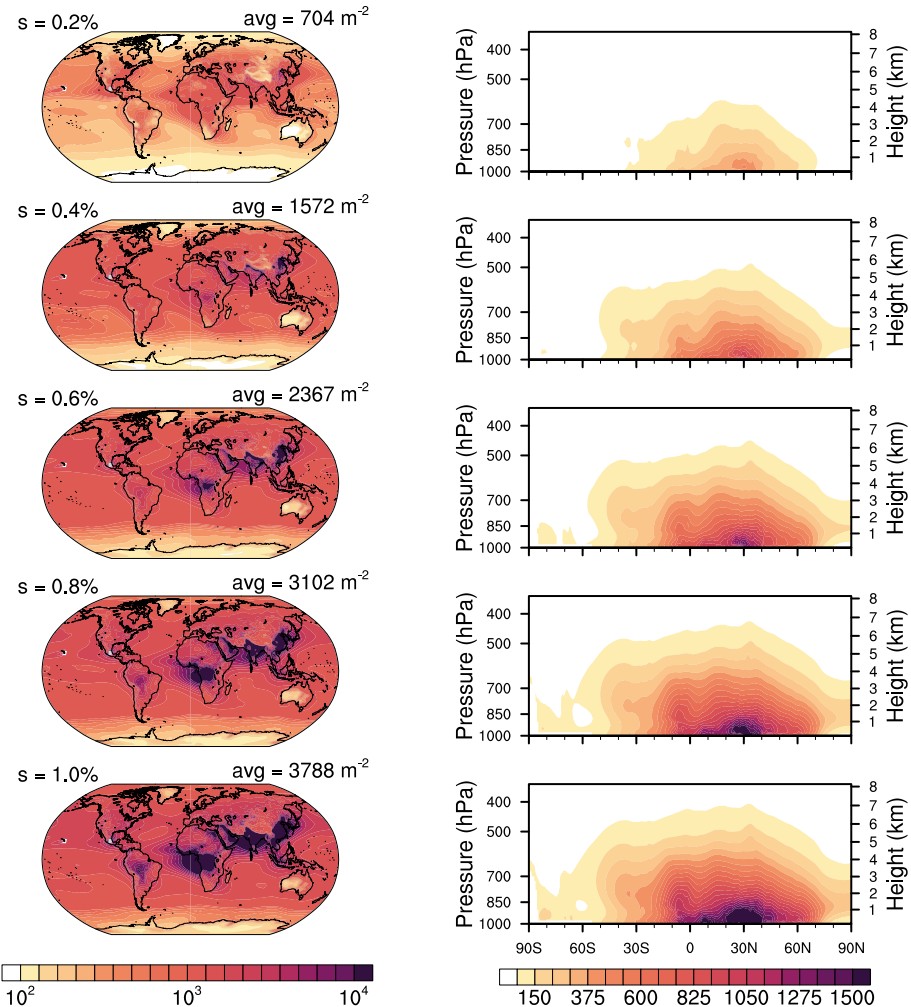

**Figure 1.** CCN load [$\times 10^9$ m$^{-2}$] of the lowermost 10 km, using vertically weighted layer depths (left panel) and corresponding zonal mean CCN concentration over height [cm$^{-3}$] (right panel) for supersaturations ranging from s = 0.2 % (top) to s = 1.0 % (bottom). CCN are averaged in time from 2003-2021, with Mauna Loa and Mexico City being excluded. Please note that the color scale for the left panel is logarithmic while it is not for the right panel.

CCN concentrations decrease towards the poles where aerosol emission is reduced. It can be seen that even though both polar regions have low aerosol and thus CCN concentrations, the Arctic is impacted from pollution advection from lower latitudes at heights of 1 to 3 km while the Antarctic is not influenced from aerosol advection over the also clean Southern Ocean. Since the assimilation of AOD is limited to the regions between 70° S and 70° N, CAMS-retrieved CCN might deviate more from observations in the polar regions since they are not constrained and we suggest to only use CCN data outside of the polar

circles. Regions with especially low aerosol loads apart from the poles are western Australia, the Weddel Sea and Greenland. Only a slight reduction of CCN by scavenging (Figure 1, right panels) can be found in the inter-tropical convergence zone (ITCZ) around $5°$ N which is determined by convective precipitation.

The general geographical distribution of global CCN concentrations and zonally averaged vertical distribution are comparable to results of other modeling studies (e.g. Lauer and Hendricks, 2006; Spracklen et al., 2008, 2011; Stier, 2016). Compared
with global retrievals from CALIPSO by Choudhury and Tesche (2022), the CAMSRA CCN also capture the general structure but the concentrations are lower with average values of near surface CCN concentrations of around 240 cm$^{-3}$ while that from CALIPSO ranges around 365 cm$^{-3}$. HoweverChoudhury and Tesche (2022) assume that dust has a non-negligible impact on CCN and thus they also find higher CCN concentrations at north-west Africa which cannot be seen to this extent in our results. This might be one of the reasons for the difference in CCN. The vertical distribution reveals that CCN abundance stays mostly
in the boundary layer and decreases with height. This agrees with the study of Choudhury and Tesche (2022) who find CCN to be predominantly located at altitudes below 2 km and decrease exponentially as the altitude increases. The larger the supersaturation becomes the more CCN are available in higher altitudes. This agrees with findings of Watson-Parris et al. (2019) who analysed the vertical distribution of global aerosol with in-situ aircraft measurements.

Figure 2 shows the zonal and meridional means of seasonally averaged CCN load at 0.2 % supersaturation. The seasonal,
longitudinal and latitudinal variation are similar with other chosen values of supersaturations (not shown). The total CCN load clearly shows a seasonal cycle, with larger loads during spring and summer within the respective hemisphere. This originates mostly from natural seasonal variations in OM (see Figure 6) considering that this is one of the main CCN contributors (see Figure 4 and 5). However, the seasonal variability is not all naturally driven, but also anthropogenically. The meridional mean CCN load reveals a strong contribution from SU (also see Figure 4 and 5) leading to high aerosol concentrations over China
($\sim 120°$ E) all-year-long. This is comparable with findings of Schmale et al. (2018) for measurements taken near Seoul in South Korea. Otherwise, we can find the seasonal variation from the top panel being reflected in the bottom panel, with higher concentrations in spring and summer originating from the OM variations over the NH landmasses. This agrees with the modeling study of Lauer and Hendricks (2006) who find the lowest CCN concentrations in the unpolluted atmosphere occur when and where biological activity is at a minimum, e.g., in wintertime or over desert areas or with very high scavenging efficiency.


Figure 3 shows the anomalies of total CCN load at 0.2 % supersaturation for different bands of latitudes. It reveals specific events when the CCN load is particularly high or low compared to the multi-year monthly mean. We have identified three of these spikes (numbered in grey) to show (1) the severe Siberian Taiga Fires in Russia in 2003, (2) the 2020 Australian Bushfires and (3) the very low CCN concentrations (especially in NH) during the summer months in 2020 and 2021, which were in parts
associated with the COVID-19 confinements in countries all over the world. These low CCN events alternate with events of high CCN abundance due to the wildfires in California in 2020 and in British Columbia in 2021.

Each fire-related spike can also be seen in the corresponding timeseries of OM and BC CCN in Figure A4. For the Australian Wildfire event there is also a strong spike in SU CCN which is an interesting feature probably occurring from pronounced DMS and SO2 production, precursor to SU, which result from a gigantic phytoplancton bloom occurring just within days of the fires

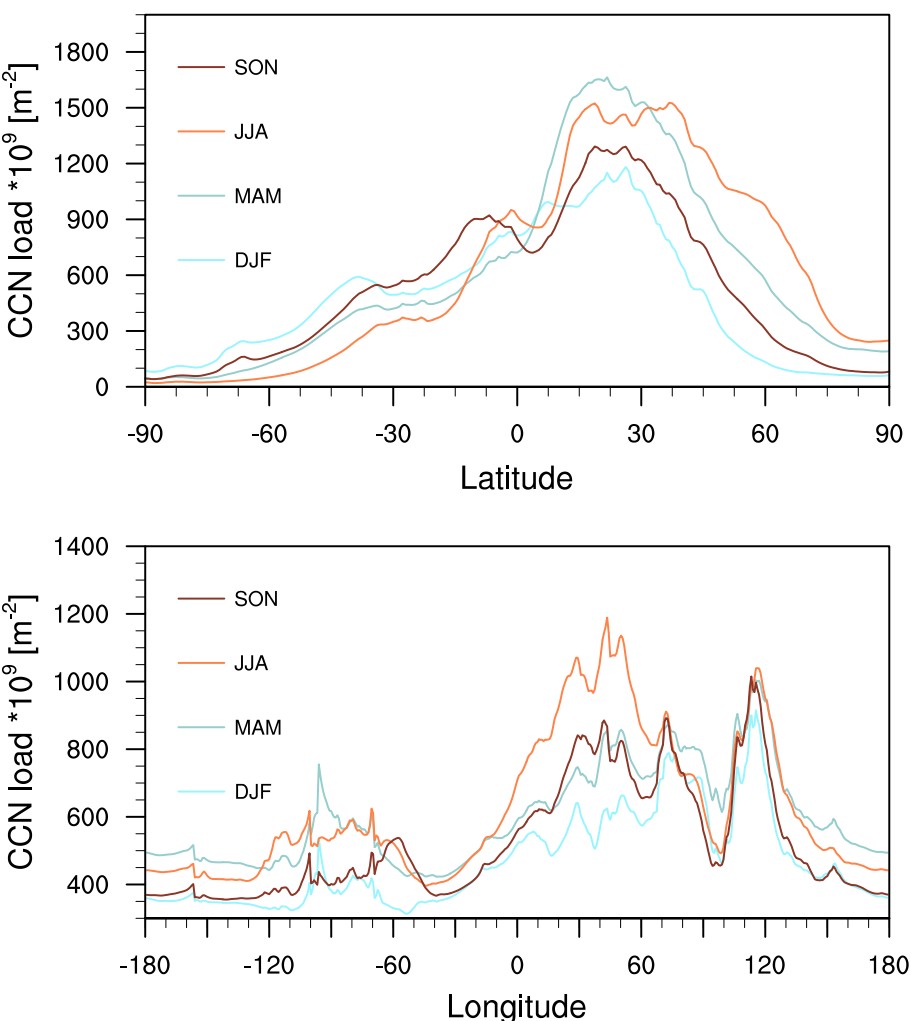

**Figure 2.** CCN load [$\times 10^9$ m$^{-2}$] of the lowermost 10 km at s = 0.2 %, averaged over longitudes (top) and over latitudes (bottom). CCN are averaged over seasons (SON, JJA, MAM, DJF) for the years 2003-2021, with Mauna Loa and Mexico City being excluded.

nourished by the depositing aerosols (Kilgour et al., 2022; Tang et al., 2021). The low CCN periods coinciding with the COVID-19 confinements are also marked by low SU CCN (see Figure A4).

We can see here that wildfires and biomass burning events show a major contribution to total CCN and that these events are well captured by the CAMS assimilation. In contrast, volcanic eruptions, e.g. the Holuhraun eruption in Iceland in 2014 (Haghighatnasab et al., 2022; Malavelle et al., 2017), are not captured by the CAMS assimilation system nor are they modelled

by the IFS as plume injections of aerosols and therefore they cannot be found in CCN load anomalies. Furthermore, we find that the time evolution of global mean CCN abundance (not plotted here) shows a significant negative trend, for s = 0.2 % it is $\approx -3 \cdot 10^9$ CCN m$^{-2}$year$^{-1}$ (on 95 % significance level), which originates from latitudes between 30° N and 60° N (here the

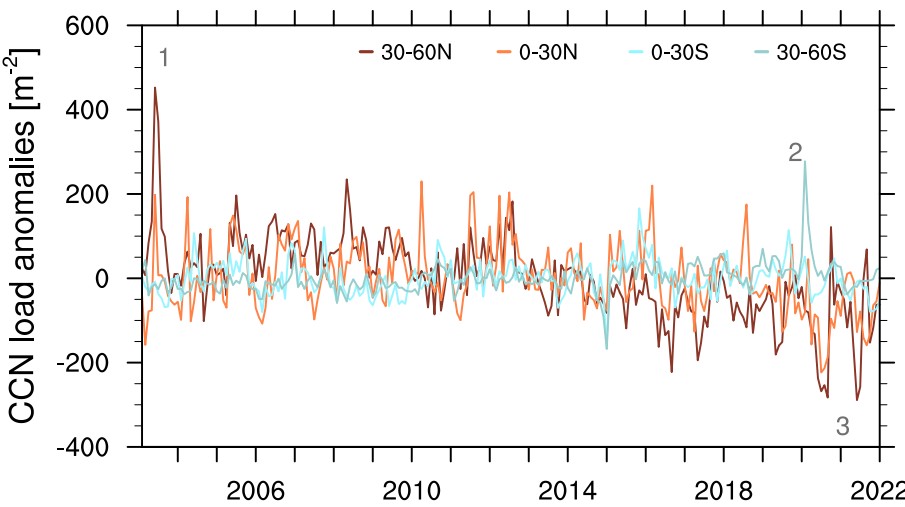

**Figure 3.** Anomalies of CCN load [$\times 10^9$ m$^{-2}$] over the entire atmospheric depth at 0.2 % supersaturation, averaged over 4 latitudinal bands between 60° S and 60° N. The monthly mean anomaly is computed by subtracting the multi-year monthly mean over the entire time-series from 2003 to 2021, with Mauna Loa and Mexico City being excluded. Numbers in grey illustrate specific showcase events.

trend is $\approx -12 \cdot 10^9$ CCN m$^{-2}$year$^{-1}$ for s = 0.2 %), while other latitudes do not show significant trends. This is reflected in trends of CCN load anomalies. The magnitude and/or occurrence of positive anomalies in the $30° - 60°$ N latitudinal band tend
to decrease, while they are increasing in absolute terms for negative anomalies as can be seen especially for the last decade in Figure 3. The values of this decrease differ for the various supersaturations but the tendencies are robust. This behavior is not surprising as anthropogenic aerosol emissions have declined in most parts of the world following air quality policies (Szopa et al., 2021). Emissions of anthropogenic primary aerosol (mostly SU) and aerosol precursors (mostly SO$_2$) have decreased in the last 20 years, and these trends are reflected in observations of aerosol abundance (Quaas et al., 2022). As illustrated in
Quaas et al. (2022) SU emissions are mostly declining since 2000, with substantial declines over North America and Europe in particular. This can be confirmed by the SU CCN component illustrated in Figure A4. According to Quaas et al. (2022) declining trends can also be found over remote oceanic regions, where ship emissions have declined since 2010 following emission control protocols. Over China, the anthropogenic aerosol emissions have been increasing until around 2010, and decreasing thereafter. Over Southeast Asia, including India, and also over parts of Africa, anthropogenic emissions show
increasing trends throughout the period 2000–2019. Organic carbon and black carbon emissions also show increasing trends which are more widespread, especially over more regions in East Asia, Africa, and also South America (Quaas et al., 2022). These general trends are mostly reflected in CAMS (Bellouin et al., 2020a) and therefore appear in CCN abundance.

### 3.2 CCN species contributions derived from CAMS

Figure 4 shows the global distributions of the four aerosol species contributing to total CCN for $s = 0.2$ % which is presented in
previous Figures. The plotted CCN are concentrations [cm$^{-3}$] near the surface, taken from the lowest model level. Additionally,

CCN for $s = 0.8\%$ are plotted in Figure A3 in the supplement. The SU CCN pattern shows a clear inter-hemispheric gradient, presenting most of the industrial and therefore SU emissions over the NH continents, with the largest emissions over China. Total CCN concentrations closely follow the distribution of sulfate since it has by far the highest concentration.

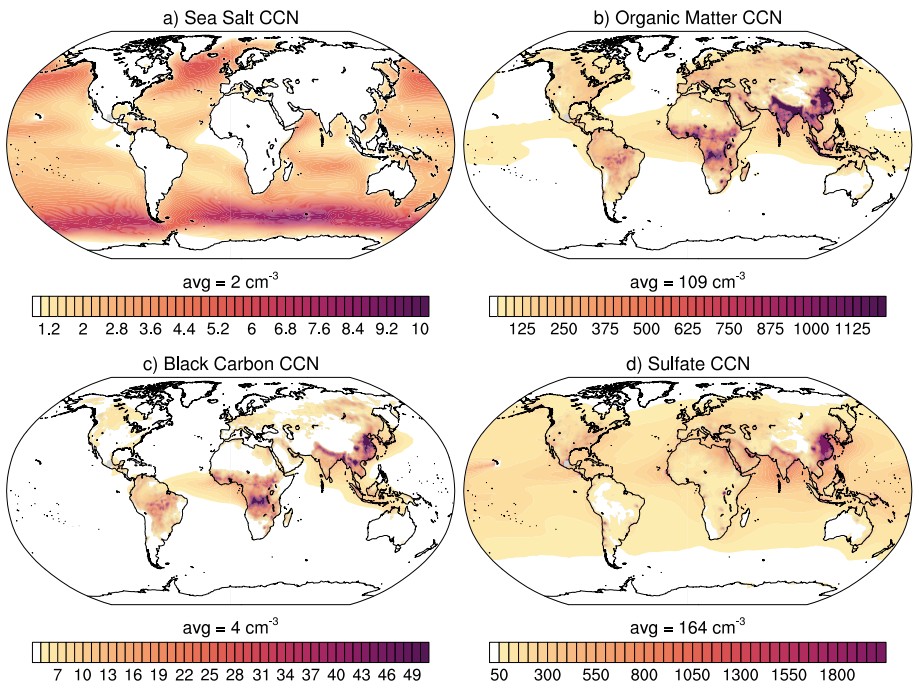

**Figure 4.** Near-surface CCN concentrations [cm$^{-3}$] at 0.2 % supersaturation of the 4 contributing species a) sea salt (sum of 3 size groups), b) hydrophilic organic matter, c) hydrophilic black carbon and d) sulfate. CCN are averaged in time from 2003-2021, with Mauna Loa and Mexico City being excluded. Please note the different scales on the color bars.

OM and BC CCN patterns are similar as both species are linked to the combustion of fossil fuel, biofuel and biomass. OM
emissions additionally originate from terrestrial and marine biogenic ecosystems. They both show high concentrations over China, northern India, and the tropical rain forests. OM has larger amounts of CCN and is more spread out than BC which, apart from additional sources, might be due to the different hydrophilic fractions at emission. Even if we would assume absolute equal and constant emission rates of BC and OM, there is still 84 % hydrophilic OM but only 20 % hydrophilic BC which can act as CCN. The largest amount of SS CCN can be found over the Southern Ocean and along the storm tracks in the North
Atlantic. SS CCN are noticeable in large storms, when surface wind speeds are high and vertical mixing is enhanced.

Figure 5 shows the contribution of the four aerosol species to the total amount of CCN at 0.2 % and at 0.8 % supersaturation. The total CCN amount is mostly divided by contributions from SU and OM. For s = 0.2 % SU dominates worldwide, except for the tropical rainforest areas and parts of Asia where OM is prevalent. At high latitudes in the northern hemisphere SU and
OM contributions often balance each other. However, for s = 0.8 % OM dominates over the continents while it balances SU

over the oceans. The contribution of OM increases with supersaturation as it consists of smaller aerosol particles than SU. Contributions from SS are negligible except over the Southern Ocean between $30°$ and $60°$ S for lower supersaturations.

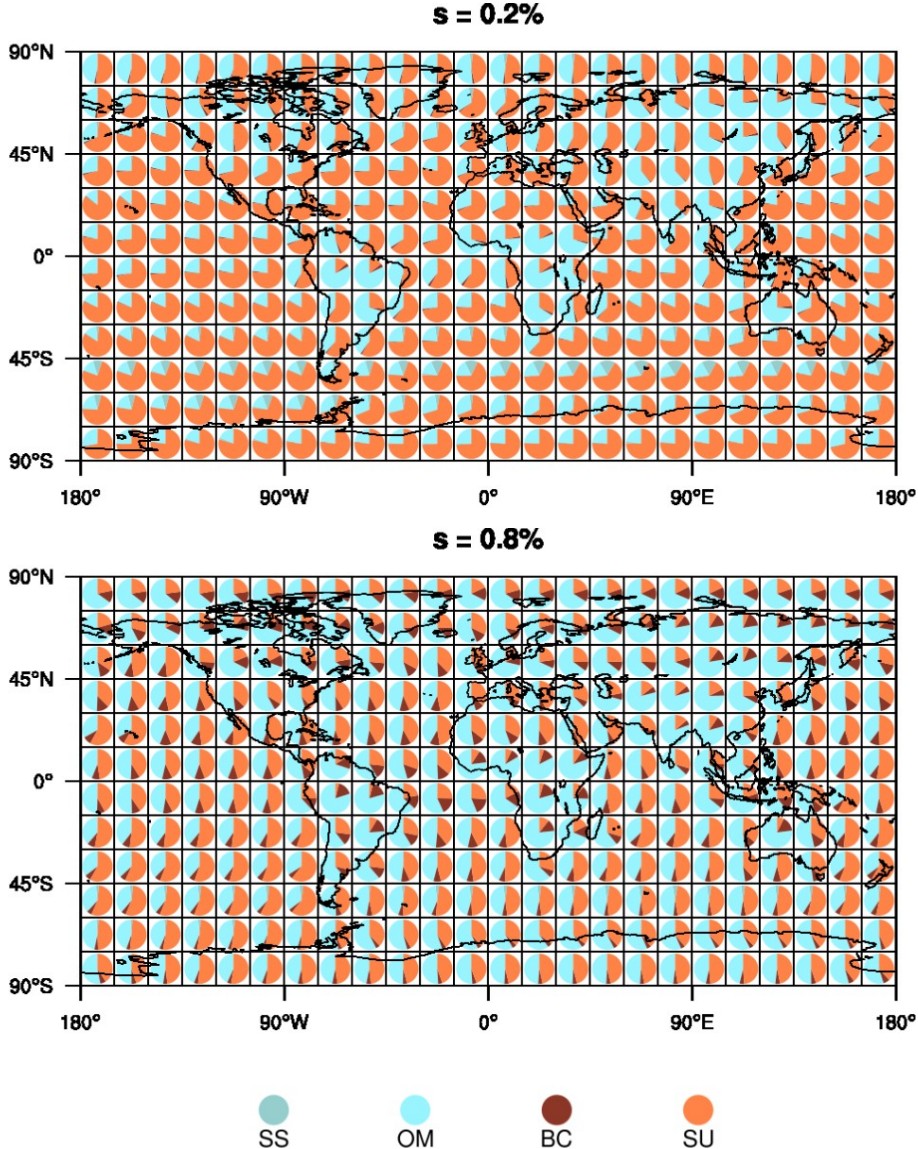

**Figure 5.** Contribution percentages of the four CCN aerosol species sea salt (SS, sum of 3 size groups), hydrophilic organic matter (OM), hydrophilic black carbon (BC) and sulfate (SU) at $s = 0.2$ % (top) and at $s = 0.8$ % (bottom) averaged in time from 2003-2021, with Mauna Loa and Mexico City being excluded. Each pie-chart is produced of a $15° \times 15°$ average from near-surface data. Please note that the pie-charts do not reflect total CCN amount.

Even though SS is highly soluble and has the largest $\kappa$ value, it is a big aerosol, associated with a large aerosol mass but low aerosol numbers. Therefore its contribution decreases when more smaller particles get activated at higher supersaturations. BC

has low contributions even in regions of heavy biomass burning, even though its contribution increases for higher supersaturation. This is due to the consideration of only 20 % being hydrophilic, and due to the smallest particle size, which requires larger supersaturations to activate in comparison to the other aerosol components.

    Figure 6 shows the variability of each CCN species over the year. For a better comparison, we have normalised the near-surface monthly mean concentrations taken from the entire time period 2003 - 2021 by dividing them by their respective time

mean. This means, values of CCN variability cannot be compared between species. In the global mean (Figure 6a), SU CCN show almost no variation over the year because NH and SH variabilities cancel each other. In each hemisphere, SU CCN are higher in the respective hemisphere's summer than in the wintertime. Because SU is a dominant contributor to total CCN, this means that total CCN loads shift between hemispheres over the year with higher concentrations in the NH during boreal summer (JJA) and higher concentrations in the SH during austral summer (DJF). Global mean OM and BC CCN seem to

peak twice during the year, however the peak in spring originates from larger emissions in the NH while the peak in autumn originates from increased emissions in the SH. SS concentrations are highest in the wintertime months over the NH and in the summer months over the SH, which is due to enhanced wind speeds in these months. In the global mean, these variations cancel each other.

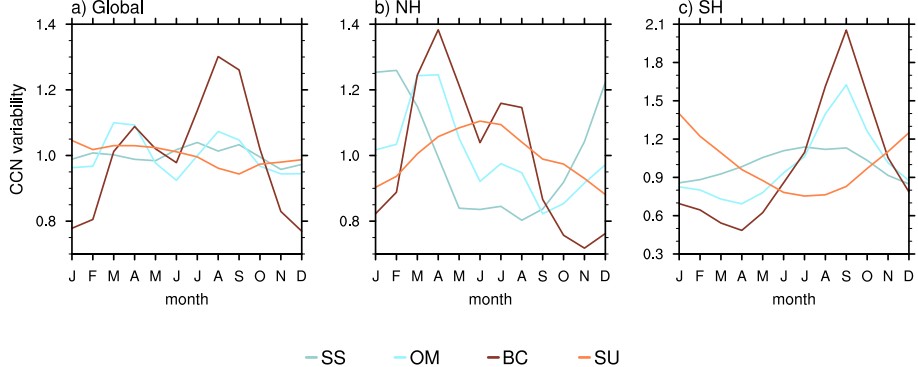

**Figure 6.** Relative variability of near-surface CCN species at 0.2 % supersaturation. Multi-year (2003-2021) monthly means of CCN concentrations are divided by the time mean to produce a normalised variability for each of the 4 contributing species: sea salt (SS, sum of 3 size groups), hydrophilic organic matter (OM), hydrophilic black carbon (BC) and sulfate (SU). Shown are averages over a) the entire globe, b) the northern hemisphere and c) the southern hemisphere, all with Mauna Loa and Mexico City being excluded.

## 3.3    CCN validation

CAMS-derived CCN are briefly evaluated here with quality-assured data from the Atmospheric Radiation Measurement (ARM) network which are available within the time frame of CAMSRA (and MACCRA, for comparison see Block (2018)) and originate from the same instrument for better comparison. The ARM surface sites used here are located in very different aerosol environments ranging from remote to highly polluted. Users are encouraged to extent this first validation with observations or retrievals of their choice. CCN concentrations are measured at several supersaturations using a Droplet Measurement Technolo-

gies (DMT) single-column CCN counter (Roberts and Nenes, 2005). The instrument steps through several supersaturations in a pyramid-like profile with 7 intervals (0.1, 0.2, 0.4, 0.6, 0.9, 1.1 and 1.2 %) in a cycle of 30 min with 5 min at each setting. The different supersaturations are obtained by variation of the chamber wall temperature and are calibrated using salt aerosol (static calibration). Additionally, supersaturations are also calculated using a heat transfer and fluid dynamics flow model (Lance et al., 2006) which are more reliable than the static calibrated ones (Shi et al., 2013). The instrument calibration and uncertainties involved are discussed in Rose et al. (2008). Spracklen et al. (2011) analysed extensive CCN observations and found a range of uncertainties from 5-40 % depending on CCN concentration, supersaturation and the type of CCN instrument used. We follow their lines and assume a relative uncertainty of $\pm 40\%$ and a minimum absolute uncertainty of $\pm 20\,\mathrm{cm}^{-3}$.

We have chosen to use the Aerosol Observing System Cloud Condensation Nuclei Average (AOSCCNAVG) value-added product (VAP) (Shi et al., 2013) because it consolidates the relevant CCN parameters into a single file and averages the data over the 5-minute integration time for each $s$ value. Since the first minute of each $s$ setting is unstable in terms of temperatures and the $s$ value overshoots the setpoint, only the last four minutes are taken into account.

**Table 2.** Atmospheric Radiation Measurement (ARM) network sites and measurement periods used for CCN evaluation (https://www. arm.gov/data/). Information on data products and quality reports can also be found on the ARM website https://www.arm.gov/capabilities/ science-data-products/vaps/aosccnavg and https://www.arm.gov/data/data-quality-program.

| Site ID | Site/Campaign | Environment | Data product | Time period (month/year) |
|---|---|---|---|---|
| SGP | Southern Great Plains: Central Facility, Lamont, OK, USA | rural | aosccnavg, level c2 | 01/2011-12/2012 |
| PVC | Cape Cod: Highland Center, Cape Cod, MA, USA | coastal | aosccnavg, level c2 | 07-12/2012 |
| PGH | Ganges Valley: ARIES Observatory, Nainital, Uttarkhand, India | mountainious | aosccnavg, level c2 | 06/2011-03/2012 |
| GRW | Graciosa Island: Azores, Portugal | marine | aosccnavg, level c2 | 04/2009-12/2010 |
| MAG | MAGIC: Los Angeles, CA to Honolulu, HI, USA - container ship Horizon Spirit | marine | aosccn100, level a1 | 10-12/2012 |

From the AOSCCNAVG VAP data product, we have chosen the c2 data level which is the highest process level. The data are taken from four land stations. Additionally we have taken AOS CCN dataset (data level a1: calibration factors applied and converted to geophysical units) from the Model CCN-100 CCN counter J. (2022) used during the Marine ARM GPCI Investigation of Clouds (MAGIC) project which is a ship campaign in the Pacific. The sites, datasets and measurement times used, are listed in Table 2.

We use daily means of quality-checked CCN measurements and ensure that each day comprises statistically stable values which can be compared to CAMS-derived CCN once a day. The comparison to CAMS CCN is done for a single supersaturation at 0.4 % for reasons of convenience. For detailed description of data handling, please see Appendix B.

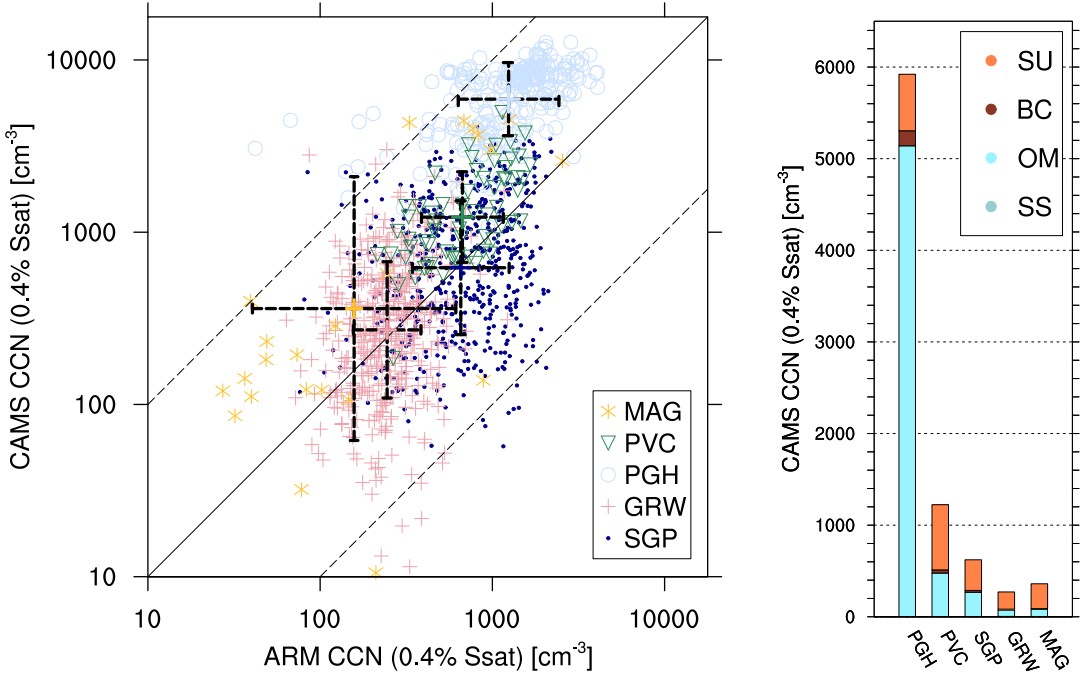

(a) CAMS vs. ARM CCN data near the surface. Average (colored cross) and standard deviations (vertical and horizontal bar) are computed from logarithmic values for each station.

(b) CAMS CCN species contributions at ARM sites, corresponding to panel (a).

**Figure 7.** Validation of CAMS CCN concentration with ARM surface site data for 0.4 % supersaturation. The ARM stations are listed in Table 2.

The evaluation of the CAMS reanalysis $CCN_{0.4}$ with ARM data (Fig. 7, (a)) reveals a good agreement considering the large range of magnitudes. As one can see directly that three out of five stations are overestimated (MAG, PVC & PGH) but a share of 96.5 % of the data lies within a factor of 10 as shown by dashed lines. As expected, the lowest CCN concentrations are found for clean marine conditions (GRW and MAG), which are followed by conditions with medium aerosol concentrations (SGP and PVC). The relative spread of data points (considering the scale of observations, see NRMSE values in Table 3) is largest for the MAG site which might be due to the low amount of data used here. The Indian site (PGH), which is found to be in one of the most polluted regions in the world, shows by far the highest model CCN concentrations and the largest deviation from observations in absolute terms. The relative spread of residuals (NRMSE, Table 3) is almost as high as for the MAG site. The CCN species contributions of the individual stations, also presented in Fig. 7, (b), reveals almost balanced contributions from SU and OM for SGP and PVC, OM dominance for PGH, and SU dominance for GRW and MAG. In comparison, SS contributions are negligible, even for marine sites.

As can be seen from Figure 8, the simulated total CCN mostly overestimate the observations in terms of median (except for SGP) and interquartile range (except for PGH). The results are summarised in Table 3. In this regard the result presented here are different from other models as many tend to underestimate observed aerosol particle and CCN number concentrations

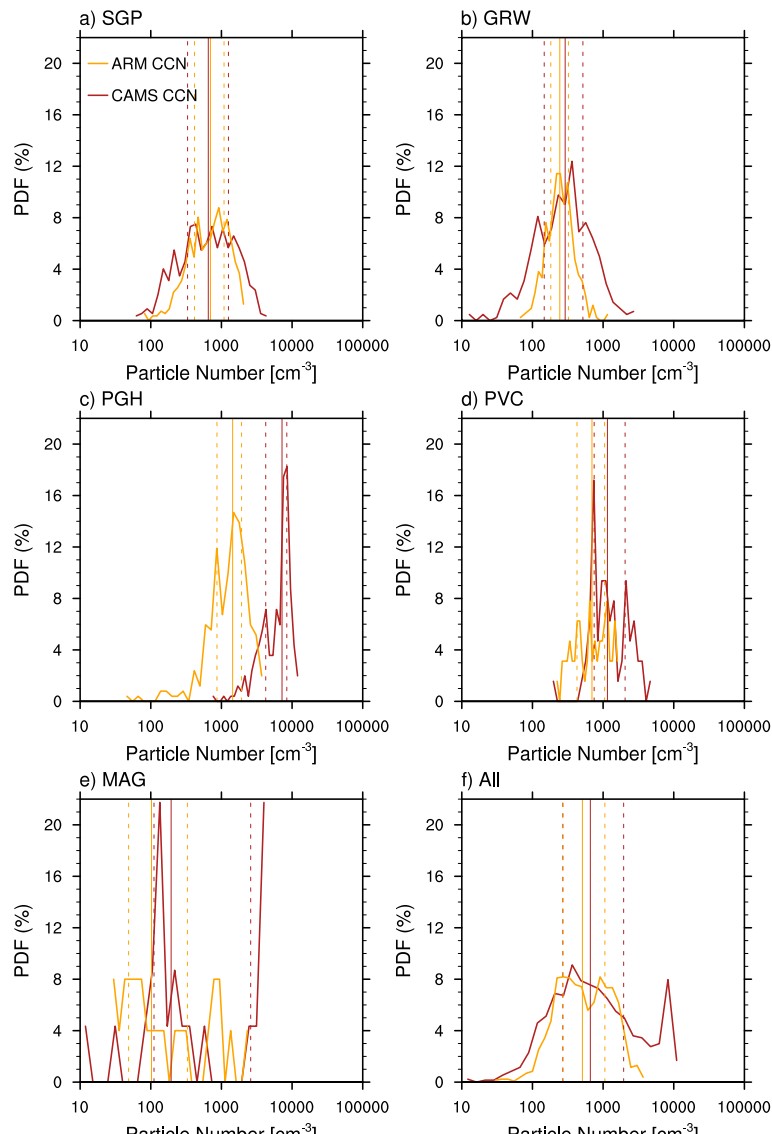

**Figure 8.** CAMS vs. ARM CCN probability density functions, for CCN taken at 0.4 % $S_{sat}$ near the surface. The PDFs are shown for the ARM sites listed in Table 2, with all stations taken together in Panel f). CAMS data are plotted in blue and ARM data in red. The vertical lines indicate the medians (solid lines) and the interquartile range (dashed), with one line at the $25^{th}$, and the other at the $75^{th}$ percentile, of each distribution.

(Fanourgakis et al., 2019). The evaluated 14 models in Fanourgakis et al. (2019) show an average normalized mean bias of all models and for all stations of -24 % and -35 % for particles with dry diameters > 50 and > 120 nm, as well as -36 % and -34 % for CCN at supersaturations of 0.2 % and 1.0 %, respectively. However, taking into account the assumed measurement uncertainties (values given in square brackets) the CCN retrieved from the CAMS reanalysis are reasonable within the range

**Table 3.** CCN evaluation with ARM data. Measurement uncertainties $\sigma_m$ in % are assumed following Spracklen et al. (2011). The probability distribution characteristics as shown in Fig. 8 are given by the median $P_{50}$, the $25^{th}$ percentile $P_{25}$ and the $75^{th}$ percentile $P_{75}$ of the CCN concentrations in [cm$^{-3}$]. The bias ratio ($P_{50}$) is the ratio of CAMS CCN $P_{50}$ to ARM $P_{50}$, regarding measurement uncertainties in square brackets, while bias ratio (IQR) is the ratio of the interquartile ranges ($P_{75}$ - $P_{25}$), respectively. A value of 1 represents the best agreement between model and observations, while values <1 show underestimation and >1 overestimation of the model. The last column indicates the normalised root mean square error (NRMSE) in % as a measure of relative deviation between model and observation where lower values indicate less residual variance. It is calculated from the root mean squared error of logarithmic values divided by $\log(P_{50})$ of the observations. The normalised RMSE is chosen to compare between samples of different scales.

| Site ID | days | data | $\sigma_m$ [%] | $P_{25}$ [cm$^{-3}$] | $P_{50}$ [cm$^{-3}$] | $P_{75}$ [cm$^{-3}$] | bias ratio ($P_{50}$) | bias ratio (IQR) | NRMSE [%] |
|---------|------|------|------|------|------|------|------|------|------|
| MAG | 23 | ARM | 40.0 | 48±20 | 102±41 | 330±132 | 1.9 [1.36,3.16] | 3.44 [1.47,8.02] | 31.7 |
|  |  | CAMS |  | 111 | 194 | 2595 |  |  |  |
| GRW | 420 | ARM | 40.0 | 182±73 | 244±98 | 323±129 | 1.19 [0.85,1.99] | 1.99 [0.85,4.64] | 16.8 |
|  |  | CAMS |  | 147 | 290 | 518 |  |  |  |
| PVC | 64 | ARM | 40.0 | 427±171 | 696±278 | 1049±420 | 1.66 [1.18,2.76] | 1.12 [0.48,2.60] | 11.8 |
|  |  | CAMS |  | 747 | 1151 | 2048 |  |  |  |
| SGP | 547 | ARM | 40.0 | 416±166 | 699±280 | 1089±436 | 0.93 [0.67,1.55] | 1.45 [0.62,3.37] | 14.9 |
|  |  | CAMS |  | 332 | 652 | 1257 |  |  |  |
| PGH | 252 | ARM | 40.0 | 865±346 | 1440±576 | 1927±771 | 5.0 [3.57,8.34] | 0.89 [0.38,2.08] | 22.8 |
|  |  | CAMS |  | 4237 | 7205 | 8421 |  |  |  |
| All | 1306 | ARM | 40.0 | 270±108 | 509±204 | 1059±424 | 1.29 [0.92,2.16] | 1.86 [0.8,4.34] | 18.1 |
|  |  | CAMS |  | 268 | 659 | 1954 |  |  |  |

of the measurements. Only the bias for PGH is out of bounds, introducing a spike in the CCN distribution which is outside the range of observations (Figure 8, f).

As stated in Inness et al. (2019b), there is a considerable change in the aerosol composition in CAMSRA compared the previous version MACCRA (Inness et al., 2013; Mangold et al., 2011; Benedetti et al., 2009; Morcrette et al., 2009). Some improvements of CAMSRA are accompanied by a large increase in OM in polluted regions from the introduction of a representation of anthropogenic SOA, while the level of SU contribution has decreased. As illustrated in Inness et al. (2019b) OM AOD has substantially increased over the entire NH compared to MACCRA, with the largest amplifications over South

West Africa (Gulf of Guinea), India and East Asia. It could be argued here, that this is the cause for the large deviation found at the PGH site, especially when compared to a similar validation of MACCRA in Block (2018) where the bias ratio for the same observations at the PGH site is only 1.28. This could hint to a possible source of bias of the CAMS reanalysis product. However, it is unclear to what extent this erroneous overestimation of CCN concerns other heavily polluted areas and it needs

further validation to find the source of this feature. Because of this, the overall bias of the log-normal distributions of all the stations taken together is $+29\%$, with an overestimation in variability by $86\%$. Note, that even if PGH showed a perfect fit (CCN divided by 5), still the overall bias overestimation would be $+21\%$ with an overestimation in variability of $23\%$. This shows, that the overestimation at PGH mainly concerns the width of the CCN number distribution.

**Table 4.** CCN-AOD correlation coefficients. CCN corresponds to CCN($s = 0.4\%$), either observed (CCN$_{ARM}$) or from the reanalysis (CCN$_{CAMS}$). AOD$_{MODIS}$ is the "Dark_Target_Deep_Blue_Optical_Depth_550_Combined" as described in Levy et al. (2013), retrieved from MODIS (collection 6 on board the Aqua), available daily on a $1 \times 1°$ grid (MYD08_D3 data product), used in here from 2003 - 2014. The Pearson correlation coefficients $R$ are computed using the logarithm of data values for which all three datasets are available. Number of days are equivalent with number of data points. The last row presents the results if all the stations are taken together as in Fig. 8, f. The values in brackets are the p-values stating the probability that the two samples have no relationship (Null-hypothesis). Values below 0.05 show that the correlation is significant at the 95% significance level.

| Site ID | days | $R$ (CCN$_{CAMS}$ vs. CCN$_{ARM}$) | $R$ (AOD$_{MODIS}$ vs. CCN$_{ARM}$) |
|---------|------|-----------------------------------|-------------------------------------|
| MAG | 15 | 0.76 (p = 0.001) | 0.65 (p = 0.010) |
| GRW | 236 | 0.18 (p = 0.005) | 0.07 (p = 0.310) |
| PVC | 43 | 0.66 (p = 0.000) | 0.44 (p = 0.003) |
| SGP | 264 | 0.24 (p = 0.000) | 0.25 (p = 0.000) |
| PGH | 189 | 0.41 (p = 0.000) | 0.28 (p = 0.000) |
| All | 747 | 0.71 (p = 0.000) | 0.37 (p = 0.000) |

The correlation coefficients (Table 4) clearly show the improvement of CAMS CCN over AOD as a proxy for CCN (Andreae, 2009), when compared to near-surface CCN measurements. This results from the resolved vertical distribution and the aerosol speciation in the CAMS reanalysis. The overall correlation coefficient considering all data points increases from 0.37 to 0.71 (almost factor 2) when using CAMS CCN instead of AOD as a proxy for observed CCN.

All correlations coefficients, except for the ones between satellite AOD and observed CCN at the GRW site, are robust at the 95% significance level as indicated by the given p-values. A study of Logan et al. (2014) shows that the Azores (GRW site) experiences a range of aerosol conditions with mixtures of dust, pollution and smoke. They found rather weak correlations between aerosol loading and CCN due to mineral dust influences, while events with sulfate content within volcanic ash and pollution particles showed strong relationship with CCN. This is also confirmed by Liu and Li (2014) who analysed the relationship between AERONET AOD and observed CCN at s = 0.4% for different ARM stations. They found weak correlations for the GRW site due to the dominant influence of sea salt and dust particles, while correlations for e.g. SGP and PGH sites were much better. Regarding their findings, one reason for the improvement of $R$ from a non-existing to at least a weak relationship at the GRW site might be related to the fact that DU is neglected here as potential CCN due to its insoluble character. For the SGP site, unfortunately no improvement could be found and the correlation values stay rather weak. For the PGH and PVC sites however we find modest improvements with correlation coefficients increasing from weak to moderate and from moderate to strong relationships, respectively. One of the advantages of CAMS-derived CCN is the vertical resolution

that enables filtering aerosol layers. e.g. from long-range transport. These layers can increase column AOD without actually increasing CCN at the relevant height (in this case near the surface), leading to lower correlations. This could be one reason for improvements we find here. The highest correlations are found for the remote marine site (MAG), where MODIS AOD is supposed to work best (Remer et al., 2005). Still, even for this site the correlation to observations by CAMS CCN is slightly enhanced in comparison to AOD even if the latter one is already strong. In general, we even find lower correlation values than Liu and Li (2014) for the ARM sites used, which might result from the use of satellite AOD rather than AERONET AOD which limits the number of retrievals.

## 4   Conclusions

The CAMS reanalysis is used to produce a 19-year-long 3-D climatology of CCN number concentrations, which is analysed and briefly evaluated here. Since the reanalysis aerosol mass mixing ratios are generated through the assimilation of satellite AOD the resulting total CCN are constrained by AOD observations from satellites. Therefore this climatology offers a unique opportunity to be used for studies of aerosol-cloud interactions in an observationally constrained global framework.

There are several advantages of using this reanalysis CCN rather than using AOD (or AI) as a CCN proxy, as was commonly done in previous observational studies of aerosol-cloud interactions. First, the reanalysis CCN have a global, spatio-temporally continuous coverage while AOD can only be retrieved for cloud-free regions at satellite overpass time. Second, the reanalysis CCN are vertically resolved while AOD is a column integrated quantity. This provides the opportunity to retrieve CCN at cloud base heights, where activation occurs. Third, the CAMS reanalysis provides mass mixing ratios of four hygroscopic and thus CCN-relevant aerosol species which are black carbon, organic matter, sulfate and sea salt. Therefore the chemical and size determined potential of each aerosol species to act as CCN is taken into account, which is not possible from AOD (or related properties such as AI) to that accuracy. Furthermore, hygroscopic growth of the aerosols is taken into account in the IFS model when computing optical properties. This reduces uncertainties associated with hygroscopic effects enhancing AOD without actually increasing CCN numbers.

The CCN climatology is available daily from 2003 to 2021, on a Gaussian grid at a resolution of 80 km and 60 vertical levels. It is derived from the CAMS reanalysis on the corresponding grid (TL255L60), using the models given parameters of aerosol lognormal size distribution and by applying a modified $\kappa$-Köhler framework. For deriving number concentrations from the given CAMS mass mixing ratios, the same aerosol size distribution for external mixtures is applied that is also used in the IFS aerosol scheme for obtaining aerosol optical properties and converting between aerosol mass and assimilated AOD. This ensures that the proportionality between CCN and AOD is kept and any improvements of derived reanalysis CCN over observed AOD can therefore only result from the vertical distribution and the modeled CCN-relevant aerosol specifications and processes.

The resulting CCN distribution shows very clearly the dependence on modeled aerosol processes, such as emission, advection and scavenging. Concentrations are pronounced in the NH mid-latitudes where anthropogenic emissions dominate. Globally, CCN are dominated by sulfate aerosol for lower supersaturations and by organic matter for higher supersaturations.

The contribution of black carbon to CCN is significantly smaller than that of sulfate and organic matter and only occurs more dominantly in connection with wildfires and biomass burning events. Contributions from sea salt are negligible except over the Southern Ocean for low supersaturations.

A brief validation with in-situ surface observations has shown, that total CCN are modeled well in the range of CCN measurements. However, we found a bias factor of 1.29, compared to the surface in-situ observations used here and an overall overestimation of the CCN spectrum. The bias is not too much considering that modeled CCN concentrations are typically within a factor of 2 – 3 of observations (Spracklen et al., 2008). The overestimation however is different from other models as many tend to underestimate CCN concentrations (Fanourgakis et al., 2019). More evaluation is necessary to confirm this.

A few limitations and uncertainties of this CCN dataset need to be summarised at this point:

- Please be aware that CCN are available once a day and not every 3 hours as the CAMSRA data product is available.

- As assimilation of AOD is limited between $70°$ S and $70°$ N the data might be less reliable at the polar regions.

- Since the species fraction is not impacted by the assimilated AOD, the total CCN amount may be more reliable than the CCN aerosol species.

- Please be aware that dust is excluded here as a potential source of CCN. This might be a point that can be adjusted in further model developments to account for observations that feature the impact of dust on CCN.

- CAMSRA points out that there is increased AOD bias around outgassing volcanoes, in particular around Mauna Loa and Altzomoni near Mexico City. Therefore we recommend to exclude these two sites as unrepresentative from the CCN dataset as was done in this study.

- Please be aware that representations of volcanic eruptions are not included in CAMSRA and thus there are no related CCN available in this dataset.

- While the modeled CCN from CAMSRA are generally high biased, we find a specific erroneous overestimation of CCN at one heavily polluted measurement site which could be related to a possible bias source in CAMSRA. It needs further validation to find the source of this feature and to investigate if this holds for other highly polluted sites.

Furthermore, it needs to be acknowledged that simplifying assumptions such as the use of non-interacting external aerosol mixtures and globally fixed aerosol size distribution and hygroscopicity parameters are used in modelling CCN from CAMSRA data. These are points which could benefit from observational constraints in future model developments. However, as stated in Paramonov et al. (2015) the total aerosol number concentration and aerosol size distribution remain more important parameters for improving the prediction of CCN than detailed information about aerosol hygroscopicity. This means that the offline diagnostic calculation of CCN as done here is very much dependent on CAMSRA and any improvements there will benefit the estimate of CCN here. For now we try to be as consistent as possible with the IFS model in terms of aerosol size distribution to ensure that the proportionality between assimilated AOD and CCN is kept.

The results show that even with the simplifications mentioned above the simulated total CCN agree well with surface observations, with a correlation coefficient of R = 0.71. In comparison to AOD with R = 0.37, the correlation coefficient significantly increases. This result shows that refining the observed column AOD by a vertical distribution and an aerosol speciation, including effects of hygroscopicity, clearly improves estimations of CCN. However, further comparisons with measurements and CCN retrievals are necessary to evaluate the modeled CCN variability and vertical distribution and to identity possible biases. The CCN climatology derived here from the CAMS reanalysis provides a new benchmark for improving assessments of aerosol-cloud interactions in particular because of its global coverage and its possible usage in Earth System Models.

## 5 Data availability

The data is freely accessible at https://doi.org/10.26050/WDCC/QUAERERE_CCNCAMS_v1 from the World Data Centre for Climate (WDCC) for registered users. When using the data, please cite as: Block, Karoline (2023). Cloud condensation nuclei (CCN) numbers derived from CAMS reanalysis EAC4 (Version 1). World Data Center for Climate (WDCC) at DKRZ. https://doi.org/10.26050/WDCC/QUAERERE_CCNCAMS_v1.

## Appendix A: The modified Köhler theory

The theory of heterogeneous droplet nucleation is founded on the work of Hilding Köhler (Köhler, 1936) who determined the equilibrium diameter of particles as a function of dry diameter $D_d$ and relative humidity $RH$ (Seinfeld and Pandis, 2006). We refer to the formulation of this relationship as the Köhler equation written as (Seinfeld and Pandis, 2006; Pruppacher and Klett, 1997)

$$\ln S = \frac{4M_w\sigma_{s/a}}{RT\rho_w D_w} - \frac{\phi_s\varepsilon\nu\rho_d M_w D_d^3}{\rho_w M_d D_w^3} = \frac{A}{D_w} - \frac{B}{D_w^3} \tag{A1}$$

with $\quad A = \dfrac{4M_w\sigma_{s/a}}{RT\rho_w} \quad$ and $\quad B = \dfrac{\phi_s\varepsilon\nu\rho_d M_w}{\rho_w M_d}D_d^3 = KD_d^3$ .

Equation A1 gives saturation ratio $S$ at a specific temperature at which a droplet is in equilibrium with its environment as a function of the wet droplet diameter $D_w$, the dry aerosol particle diameter $D_d$ and the hygroscopicity of the aerosol particle. The curvature parameter A contains the surface tension parameter $\sigma_{s/a}$, the universal gas constant $R$, temperature $T$, the molar mass of water $M_w$ and the water density $\rho_w$. It shows that at any given temperature the equilibrium vapor pressure over a curved interface exceeds that of the same substance over a flat surface. The difference in equilibrium vapor pressure from a curved to a flat solution increases the smaller the droplet becomes (Seinfeld and Pandis, 2006).

The hygroscopicity parameter B contains the mass fraction of soluble material $\varepsilon$, the osmotic coefficient $\phi_s$, the number of ions $\nu$ that a molecule dissociated into when dissolved in water, the dry aerosol density $\rho_d$ and aerosol molar mass $M_d$, the molar mass of water $M_w$ and the water density $\rho_w$ as well as the dry aerosol particle diameter $D_d$.

The B parameter cannot be easily obtained and is very complex. In order to characterise the relative hygroscopicities of individual aerosol constituents, known mixtures, and complex atmospheric aerosols, Petters and Kreidenweis (2007) introduced a hygroscopicity parameter $\kappa$ which simplifies the hygroscopicity parameter $B$ and obviates the need to determine, or assume, aerosol properties such as dry particle density, molecular weight, and dissociation constants (Petters and Kreidenweis, 2007):

$$S = \alpha_w \cdot \exp\left(\frac{4M_w\sigma_{s/a}}{RT\rho_w D_w}\right) \quad \text{with} \quad \frac{1}{\alpha_w} = 1 + \kappa\frac{V_d}{V_w} \tag{A2}$$

with $\kappa$ being defined through its effect on the water activity of the solution $\alpha_w$, $V_d$ being the volume of the dry particulate matter and $V_w$ being the volume of the water. With some further reconstructions (see Petters and Kreidenweis (2007)), this evolves into Eq. 3, stated here again as

$$S(D_w) = \frac{D_w^3 - D_d^3}{D_w^3 - D_d^3(1-\kappa)} \cdot \exp\left(\frac{4\sigma_{s/a}M_w}{RT\rho_w D_w}\right) \text{with} \quad \alpha_w = \frac{D_w^3 - D_d^3}{D_w^3 - D_d^3(1-\kappa)} \ . \tag{A3}$$

Now, we modify this equation as in Pöhlker et al. (2023). Assuming supersaturations $\Delta S \ll 1$, we can approximate $\ln S = \ln(1 + \Delta S) \approx \Delta S$, and with assuming $D_d \ll D_w$, it becomes

$$\ln S \approx \Delta S = \ln\left(\frac{D_w^3 - D_d^3}{D_w^3 - D_d^3(1-\kappa)}\right) + \left(\frac{4M_w\sigma_{s/a}}{RT\rho_w D_w}\right) = \ln\left(\frac{D_w^3 - D_d^3}{D_w^3 - D_d^3(1-\kappa)}\right) + \frac{A}{D_w} \tag{A4}$$

$$\begin{aligned}
\Delta S &= \frac{A}{D_w} - \ln\left(\frac{D_w^3 - (1-\kappa)D_d^3}{D_w^3 - D_d^3}\right) \\
&= \frac{A}{D_w} - \ln\left(1 + \frac{\kappa D_d^3}{D_w^3 - D_d^3}\right) \\
&\approx \frac{A}{D_w} - \frac{\kappa D_d^3}{D_w^3 - D_d^3} \\
&\approx \frac{A}{D_w} - \kappa\frac{D_d^3}{D_w^3}
\end{aligned} \tag{A5}$$

As we can see, even for our set conditions of low supersaturations, this relation holds according to the traditional Köhler formulation, but with $K = \kappa$.

## Appendix B: ARM data treatment

CCN evaluation with ARM data is done for daily means of quality-checked data. Wood et al. (2017) found that for the GRW data, there was an abnormal degradation in CCN concentrations from October 2009 to June 2010. The values returned back to normal (in comparison to concentrations from another condensation nuclei (CN) counter, where CN refers to concentration

of all particles greater than approximately 10 nm in diameter) after the instrument was maintained thoroughly. They corrected the data using monthly multiplication factors to obtain a stable ratio between CCN and CN, assuming that the CN counter was correct. In this study we also use the corrected dataset from Wood et al. (2017).

Special care is taken for the daily mean statistics, which is used to compute CCN at 0.4 % $s$. To ensure a statistically stable result, only CCN data retrieved at at least 4 of the 7 $s$-bins with a minimum of total 96 measurements per day (1/3 of maximum possible data coverage) are taken into account. Further we neglect data which seem to have artefacts like a systematic significant increase of total number concentration with supersaturation. This is mainly found for GRW data. Therefore, the corrected dataset from Wood et al. (2017) is only applied on days with good daily statistics.

The comparison to CAMS CCN is done for a single supersaturation at 0.4 % for reasons of convenience. Are there enough data at 0.4 % $s$ available, the daily average is simply taken from those measurements. Otherwise, the measured data from the various $s$-bins are converted to $CCN_{0.4}$ as is done in Andreae (2009), using Twomey's power law (Twomey, 1959; Seinfeld and Pandis, 2006)

$$CCN(s) = CCN(s = 1\%) \cdot s^k, \tag{B1}$$

with supersaturation $s$ in %. Solving Equation B1 for k gives

$$k = \frac{1}{\ln s} \cdot \ln\left(\frac{CCN_s}{CCN_{1\%}}\right) \tag{B2}$$

and extending it to a more general form considering CCN at two different supersaturations, regarding that $\ln(s = 1\%) = 0$,

$$\ln CCN_2 - \ln CCN_1 = k \cdot (\ln s_2 - \ln s_1), \tag{B3}$$

one obtains this simple form

$$\ln CCN_2 = k \cdot \ln\left(\frac{s_2}{s_1}\right) + \ln CCN_1. \tag{B4}$$

where $CCN_1$ is the given number concentration of CCN at supersaturation $s_1$ and $CCN_2$ is the desired number concentration of CCN at supersaturation $s_2$. Taking the exponential of Equation B4, we obtain the final form, which is used in this study to convert CCN at any measured $s$ to CCN at s = 0.4 %

$$CCN_{0.4} = CCN(s) \cdot \left(\frac{0.4}{s}\right)^k. \tag{B5}$$

The exponent k is computed from linear regression between logarithms of $s$ and respective CCN close to the desired value of s. The actual behaviour of CCN with $s$ is not exactly following the power law. A demonstration and a possible extension of the formular are given in Cohard et al. (1998). This deviation is accounted for by the standard deviation of k. But since this only adds about 1-3 % uncertainty to $CCN_{0.4}$, it can be neglected compared to the 40 % measurement uncertainty.

**Appendix C: Supplementary Figures**

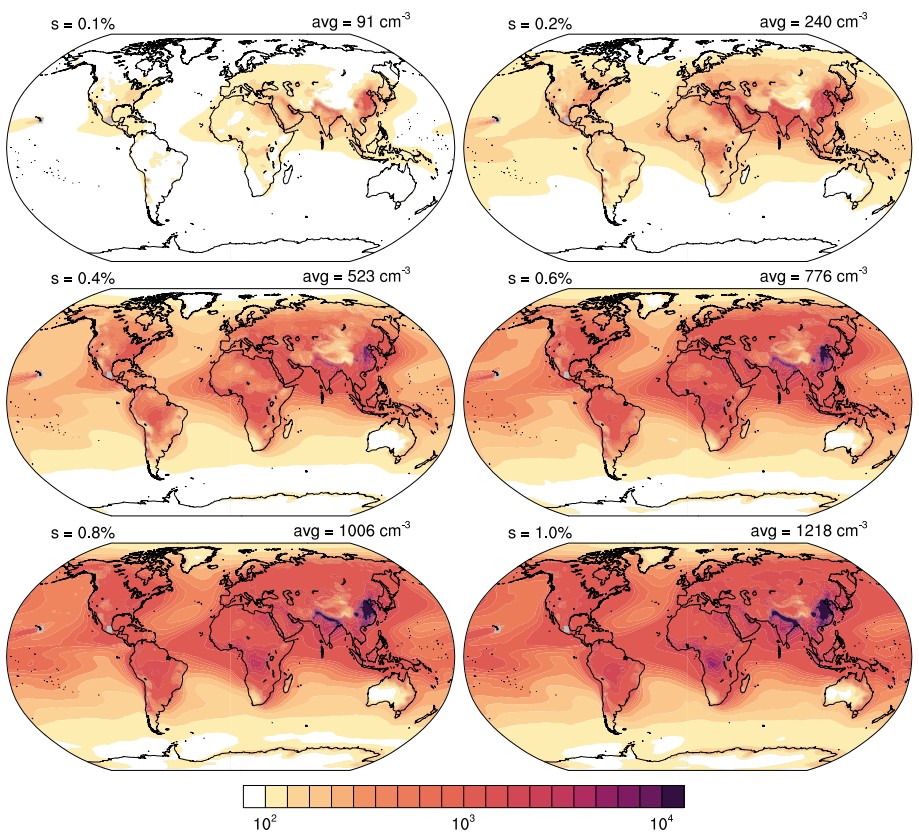

**Figure A1.** CCN concentrations [cm$^{-3}$] averaged over the lowermost 1 km above surface, using vertically weighted layer depths, for super-saturations ranging from s = 0.1 % to s = 1.0 %. CCN are averaged in time from 2003-2021, with Mauna Loa and Mexico City being excluded.

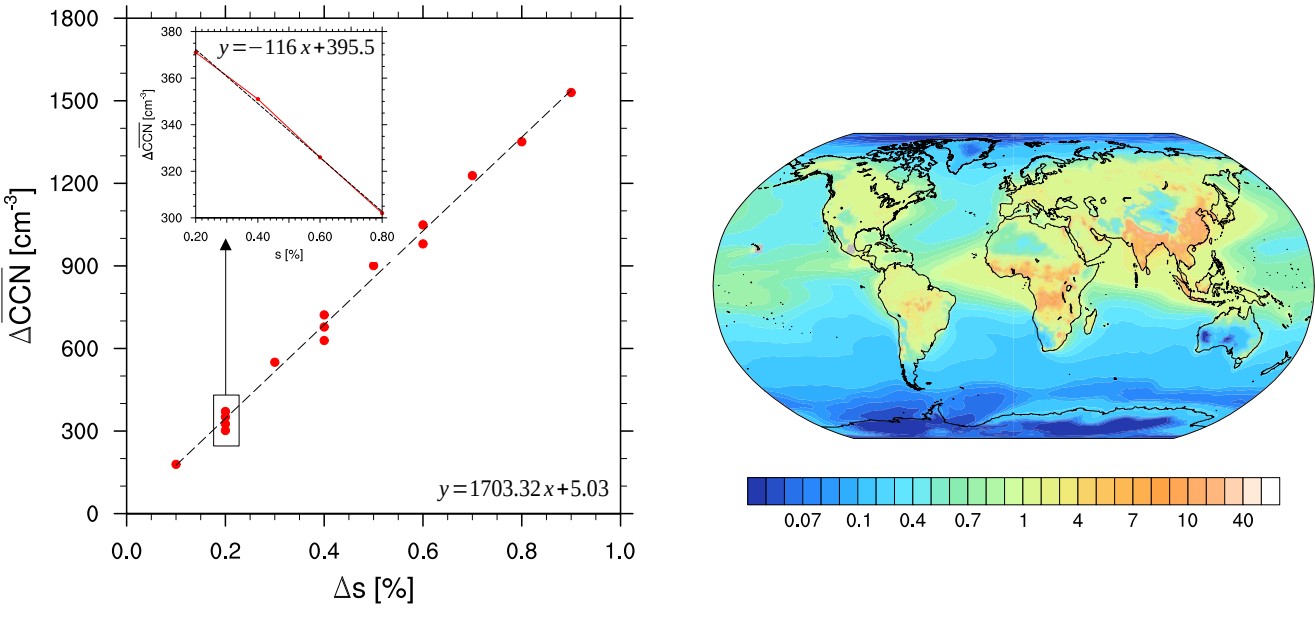

a) Change of global mean CCN [cm$^{-3}$] with change of s [%]

b) Normalised $\Delta$ CCN for a 0.1%-point increase in s.

**Figure A2.** The left panel (a) shows the change of CCN [cm$^{-3}$] with change of supersaturation $s$ [%]. The values are obtained from spatially-weighted global means of CCN number concentrations in the lowest model level. The deviations of $\Delta$ CCN at $\Delta s = 0.2\%$ are plotted again for those supersaturations which $\Delta s$ is added to. The black dashed line represents the linear regression for which the function is given in the figures. The right panel (b) shows the normalised CCN difference for a 0.1% increase in supersaturation. The local change in CCN is divided by the global mean difference, thus the relative change is presented.

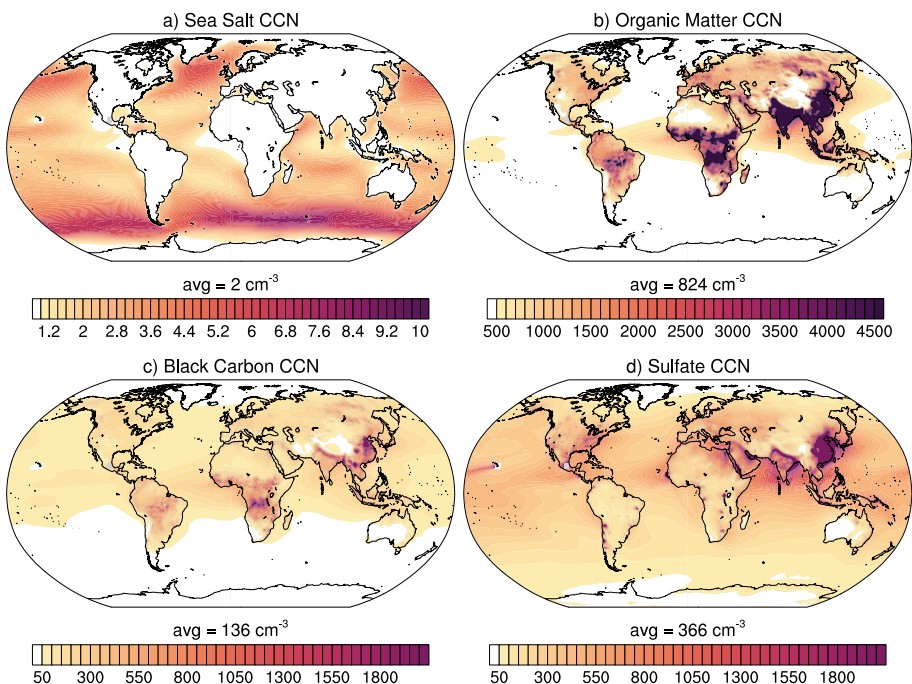

**Figure A3.** Near-surface CCN concentrations [cm$^{-3}$] at 0.8 % supersaturation of the 4 contributing species a) sea salt (sum of 3 size groups), b) hydrophilic organic matter, c) hydrophilic black carbon and d) sulfate. CCN are averaged in time from 2003-2021, with Mauna Loa and Mexico City being excluded. Please note the different scales on the color bars.

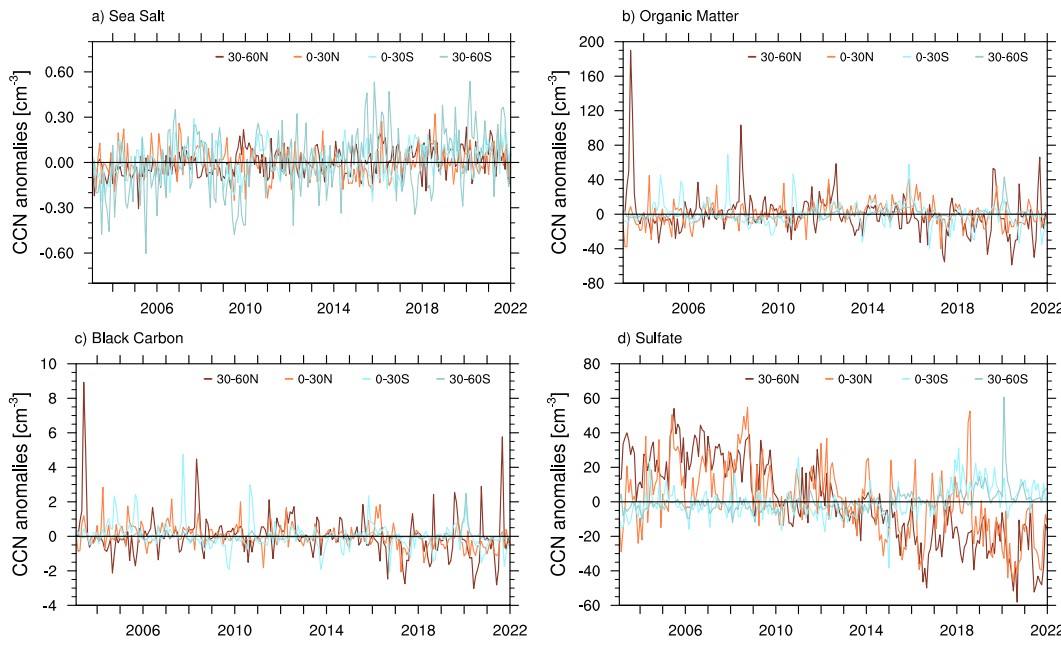

**Figure A4.** Anomalies of near-surface CCN [cm$^{-3}$] at 0.2 % supersaturation, averaged over 4 latitudinal bands between $60°$ S and $60°$ N. The monthly mean anomaly is computed by subtracting the multi-year monthly mean over the entire time-series from 2003 to 2021, with Mauna Loa and Mexico City being excluded.

*Author contributions.* Data analysis, including coding the box model, running the code on CAMS data, validation and ARM data handling, creating statistical analyses, plots, tables and text by main author Karoline Block. Help in setting up the initial box model version by Daniel Partridge. Downloading and handling CAMS data, helping with the analysis and bug fixes in the box model by Mahnoosh Haghighatnasab. Project advise, valuable input and proof read by Philip Stier. Project lead, corrections and proof read by Johannes Quaas.

*Competing interests.* The authors declare that they have no conflict of interest.

*Acknowledgements.* This study is supported by the European Union through the Horizon2020 project MACC-III (grant agreement 633080), the COPERNICUS Atmospheric Monitoring Service (CAMS-74, radiative forcing), European Research Council (ERC) Starting Grant QUAERERE (GA 306284), and Horizon 2020 project CONSTRAIN (GA 820829).

The CCN dataset produced for this study is generated using Copernicus Atmosphere Monitoring Service information (2003 - 2021). Neither the European Commission nor ECMWF is responsible for any use that may be made of the Copernicus information or data it contains. The CAMS data are downloaded from the Copernicus Atmosphere Monitoring Service (CAMS) Atmosphere Data Store (ADS) (https://ads.atmosphere.copernicus.eu/cdsapp#!/dataset/cams-global-reanalysis-eac4?tab=overview).

ARM data are obtained from the Atmospheric Radiation Measurement (ARM) user facility (https://www.arm.gov/data/), a U.S. Department Of Energy (DOE) Office of Science user facility managed by the Biological and Environmental Research Program. Data correction for the GRW site was kindly provided by Robert Wood.

The MODIS products used in this study are from the NASA Goddard Space Flight Center and available from the Atmosphere Archive and Distribution System (http://ladsweb.nascom.nasa.gov).

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
