# Peer review of "Cloud condensation nuclei concentrations derived from the CAMS reanalysis"

_Earth System Science Data, 2023_

## Author Comment (AC1)

Reply to comments by Anonymous Reviewer#1

This manuscript introduces a new, comprehensive atmospheric CCN data set based on aerosol reanalysis, and shortly investigates the performance of this proxy against observations and compared with an earlier proxy. The new CCN product introduced in the paper is highly relevant for aerosol-cloud climate investigations, and is anticipated to have a wide range of applications. This paper is scientifically robust, and very well written and structured. I have only a few, relatively minor, issues that should be fixed before accepting this paper for publication.

We thank the reviewer for reading the manuscript carefully and acknowledging the usefulness of the new CCN dataset.

The manuscript provides a valuable discussion on the problems associated with CCN estimated from AOD measurements (section 1). Related to this, although the new CCN proxy shows major improvements over the AOD-derived one, it is still far from perfect, as can be seen from Figures 3 and 4 and Table 3. The authors should better acknowledge the remaining uncertainties, rather than claiming a good agreement with measurements (lines 378-379).

We have rewritten this paragraph and added more information regarding the uncertainties of the model compared to the measurements.

The paper would benefit from a bit more detailed discussion (1-2 paragraphs) on what might cause the remaining problems with the new CCN proxy (see my previous comment), and what could be done to improve the proxy further. One clear issue related to this is the assumption of an external aerosol mixture in the aerosol reanalysis product. This poses certainly challenges for estimating CCN concentrations, as many of the simulated components (especially SU, OM and BC) are closer to internal than external mixtures in large parts of the atmosphere.

We have added a paragraph in the conclusions that summarises the limitations of the new CCN dataset and also included more about remaining uncertainties.

However, we cannot clearly say what causes deviations of the modeled CCN from measurements as there are a lot of criterias and assumptions just from the model site. It would not be valid to say that the proxy would be better if internal mixtures were assumed, even if they are closer to reality. It comes down to how the internal mixtures are described in their size distributions, how well interactions are parameterised. Another big source of uncertainty is the emission scheme as CCN depend very much on the availability of aerosols. This is something we cannot change in our model that computes CCN as we are dependent on the IFS model and the assimilation scheme used in CAMS.

In this regard, we have one specific hint about one specific location (PGH site) that we discuss further in the document.

Saying that the correlation between the CCN proxy and measured CCN concentration almost doubles when using the new CAMS-derived CCN proxy compared with AOD-derived CCN proxy (lines 400, 416 and 450) is not a statistically robust statement. For example, doubling the correlation from 0.49 to 0.98 would be a huge improvement, while doubling it from 0.01 to 0.02 would not help at all in practice. Please reword using statistically relevant measures in expressing the improvement in the new proxy compared with the old one.

Yes, we rephrased this part.

Finally, I do not understand what is meant by the last sentence in section 3.2 (lines 350-351).

Yes, we have deleted this sentence, as we actually do not analyse anthropogenic contribution as it is implicit to the dataset.

---

## Author Comment (AC2)

Reply to comments by Anonymous Reviewer#2

This study employs the CAMS reanalysis aerosol mass mixing ratios from various aerosol species to calculate CCN number concentrations using estimated parameters of the aerosol particle size distribution. The results are provided globally over a 19-year period, and various spatial distributions, time series, and a brief comparison with observations are presented. In my opinion, this manuscript is well-written, and the analyses within it are presented clearly. This will be a great paper and the community will benefit substantially from this study. Therefore, I recommend publication after addressing the comments below.

We thank the reviewer for reading the manuscript carefully and for giving such constructive and detailed comments! That really helped us to improve this manuscript. We have altered the manuscript quite a bit according to your suggestions and hope that it is now better understandable.

General Comments:

Your methodology is explained clearly. It would be great to elaborate on the exclusion of dust particles from the CCN analysis.
- While your study excludes dust from CCN calculations, it's worth noting that many studies consider dust to be an important contributor to CCN concentrations. For instance, Che et al. (2022) employed UKESM to investigate CCN sources, and they included dust in their analysis.
- They highlighted that even though dust particles are generally insoluble, wettable dust particles with larger diameters can function as CCN. Moreover, small dust particles can accumulate soluble materials through internal mixing during transport, significantly enhancing their activation potential (Bègue et al., 2015; Dusek et al., 2006; Gibson et al., 2007; Hatch et al., 2008).
- This significance of Saharan dust as CCN at a 0.2% supersaturation has also been emphasized in studies by Weinzierl et al. (2017) (Figure 12) and Haarig et al. (2019) (Figure 3).
While you've provided an explanation about the treatment of dust in CAMS in Section two, it is excluded from the results in Section three. I know you provided a brief explanation about dust as CCN towards the end of Section 3.3, however you could elaborate on your justification for excluding dust particles, ideally accompanied by supporting citations. Additionally, I recommend relocating this discussion to Section two, potentially integrating it at the end of 2.1 or the beginning of 2.2.

We have included a paragraph in section 2.2 where we now elaborate on the exclusion of dust.

It would be good to compare your results with previous studies and highlight the similarities and differences. In particular, Sections 3.1 and 3.2 (maps, time-series, etc.) do not provide any observations. I've added some examples in the specific comment part, but I encourage you to add any other helpful references with explanations.

We have included several references comparing the spatial structure of CCN with modeled estimates and recent retrievals using Calipso Lidar. We also included more references in the validation section looking at measured CCN. For the time series analysis we already have compared the features with satellite retrievals of aerosol abundance, but extended the analysis for the various aerosol components.

Given the significant influence of CCN on low clouds, have you examined the figures for the lower troposphere or surface, rather than the whole troposphere? I really don't want to impose additional analysis, but if you already have those figures, it would be ideal to include them in the supplementary.

We focused on the CCN load first to not miss any features happening at higher altitude levels. But the CCN near the surface, be it just one the lowest model level or averaged over a certain height range (see now Figure C1), shows the same features as the CCN load.

Figures are very informative. Many figures are based on 2007 averages. Why not use the 19-year averages? If you plotted figures with the full climatology, you could add show them in the paper, or add a few of them to the supplementary.

We have chosen to only use one year (arbitrarily) because it already shows all the features we wanted to highlight. However, we agree that it might be more informative to show the 19-year average and add informative numbers on the plots . This is changed for all Figures. Nevertheless, the conclusions stay the same.

Most figures and tables are based on one supersaturation (0.2% or 4%). Can you add a brief explanation on 1) why 0.2% is preferrable and 2) how different the results are when using other supersaturation values.

We have chosen to show results for a rather low supersaturation for some of the plots because this is an often used value in other studies (e.g.  Choudhury & Tesche 2023, Stier 2016, Spracklen 2011, Schmale et al 2018)  and typically reached in stratocumulus updraught cells. However, it is not a value that is preferable at any circumstances as it is always case-dependend. Figure 1 showcases the differences in load and extent of CCN activation for various supersaturations. We have now added Figure C2 in the Appendix, illustrating the change of CCN with supersaturation, and added a little extra description in the text.

The abstract could benefit from a more specific description of the methods and results. I've provided a few suggestions in the specific comments part.

Ok, we have changed the abstract according to the specific comments

If you haven't done so already, please provide a description of limitations on your dataset website. For instance, mention the possible biased values over polar regions, as CAMS is not constrained in polar regions (as you mentioned in L413); or the AOD assimilation to CAMS calculation of mass mixing ratio is less reliable over bright surfaces or for cloudy conditions.

This data description paper (when published) is linked to the dataset website. We think it will be enough to refer to what is written in this paper. However, we will summarize all limitations more comprehensively in the document so the reader does not need to search for the information.

The Conclusions part could benefit from summarizing the limitations of your study. Following up on the previous comment, the limitations in the polar region should be listed in a paragraph in the conclusions and possibly in the abstract. You can mention a note in the abstract, for example, "The derived CCN based on CAMS is more reliable over the tropics and mid-latitudes."

We have summarised the limitations in the conclusions and mention parts of it in the abstract as well.

Also, the issue with the polar region needs to be mentioned earlier when describing Figure 1.

We have included a comment here as well

Since CAMS outputs are provided every 3 hours, can you comment on 3-hourly CCN? Do you expect less agreement with observations? I understand that 3-hourly data can take up so much space. Therefore, if you plan to continue this work in the future, it might be helpful to consider a daily maximum dataset in addition to daily mean data.

Thank you for this suggestion, this would be indeed helpful for later developments. A higher temporal resolution of CCN would of course be beneficial for observation comparison and for input in cloud resolving models. In fact, there is work going on looking into this in a modeling framework. However, on longer time scales (months and years) and considering global coverage, the daily variability of CCN

is not as important. Having CCN once a day is enough to capture the general distribution in time and space.

Equations and derivations are described clearly. I encourage the authors to check again and add units for all variables (The units are missing for some variables). This will help better understand the variables and will make a huge difference for scientists who want to implement the derivations into their codes/models.
ok

There are many occurrences of the word "concentration," but often it is not mentioned whether it is about number concentration or mass concentration. You can add a note at the beginning of the paper clarifying that you mean number concentration unless stated otherwise (If that is true).
Yes, we added a comment in the beginning of the introduction

Are spatial averages weighted by the area of each grid cell? This can be mentioned in the paper.
Yes, spatial averages account for the size of the grid cells and vertical averages and loads are weighted by the vertical layer thickness.

Specific Comments:

L1 and 16: Perhaps change "concentrations" to "number concentration."
ok

L13-15: The results of your study need to be mentioned more specifically. Instead of the first sentence, maybe mention that the derived CCN dataset captures the general trend and spatial and temporal distribution of total CCN and CCN from different aerosol species. For the second sentence, it would be good to add the correlation values, as they have been improved.
ok

L22-26: I encourage you to add references for each sentence.
We have added some more references here

L39-40: Although AOD and AI do not inform about the aerosol species, the angstrom exponent is related to aerosol size and therefore implies aerosol type. It would be good to mention this, along with relevant references, e.g., Kapustin et al., (2006) and Shinozuka et al. (2015) (Figure 3).
We now elaborate a bit more on the use of AI

L40: I think the use of "this study" should be reserved for instances where you refer to your current manuscript. Please mention the exact reference to avoid confusion.
ok

L53-68: Is this about the surface or vertical distribution of aerosol properties?
Most of the studies mentioned in this paragraph focus on spatial distributions, variability in time or co-variation of parameters for a specific location or region. This can include a vertical column integral, depending on the parameter. However, vertical distributions of aerosol properties are studied e.g. by Winker et al. (2013) or by Choudhury and Tesche (2022), as is mentioned in this manuscript.

L61: Are the "various techniques" here in-situ measurements, ground-based remote sensing, or satellite remote sensing?

We are talking here of in-situ measurements. We have altered this sentence to "In-situ measurements of aerosol mass concentrations, which were conducted at 24 European sites by various institutes using different instruments and techniques, have been comprehensively assessed in a European aerosol phenomenology (Van Dingenen et al., 2004; Putaud et al., 2004, 2010) which summarizes PM10 and PM2.5 mass concentrations, their chemical composition and aerosol particle size distributions."

L78-80: Please mention if this is 3D or just at the surface.

These are ground-based samples.

L85-87: Perhaps a rephrasing is needed, as it is not clear if GCMs and observations are compared, or if different observational methods are compared.

We deleted this sentence

Paragraph starting at L99: Please add the temporal and spatial resolutions of the CAMS model. Additionally, does CAMS assimilate meteorological data in addition to satellite observations?

Yes it does. We have added some lines here.

L115-120: Please add references.

Ok

L153-156: Can you be more specific about the cloud properties in CAMS? Does CAMS have a cloud microphysics model coupled to the aerosol model, or does it receive cloud parameters from another model/reanalysis?

Cloud properties are modeled with a detailed 1-moment cloud microphysics scheme in the IFS model[1,3]. It is not interactively coupled to the aerosol model (other than in the radiation scheme!, see Innes et al., 2019b), but the aerosol mass concentrations are parameterized as climatological means[2]. Experimental coupling has been conducted[4]. We added a comment here in the manuscript.

Cloud properties are somewhat constrained (as in ERA5 [5]), however experiments on assimilating space-born cloud radar and lidar observations for cloud profiling are ongoing[6].

References:
(1) IFS CY48R1 model documentation: https://www.ecmwf.int/en/elibrary/81370-ifs-documentation-cy48r1-part-iv-physical-processes

(2) IFS aerosol climatology: https://www.ecmwf.int/en/elibrary/80028-implementation-cams-based-aerosol-climatology-ifs

(3) IFS cloud microphysics: file:///home/kblock/Downloads/9441-new-prognostic-bulk-microphysics-scheme-ifs.pdf

(4) IFS ACI: https://www.ecmwf.int/sites/default/files/elibrary/2011/11283-aerosol-cloud-radiation-interactions-and-their-impact-ecmwfmacc-forecasts.pdf

(5) ERA5 global reanalysis: https://rmets.onlinelibrary.wiley.com/doi/epdf/10.1002/qj.3803

(6) Cloud assimilation: https://www.ecmwf.int/en/newsletter/162/meteorology/progress-towards-assimilating-cloud-radar-and-lidar-observations

L165-167: MODIS has an AOD product called deep-blue (DB) that covers over bright surfaces where dark-target (DT) AOD is missing. It seems that CAMS model does not use DB because it was not available when assimilation of DT was implemented in CAMS. It might be good to mention a possible solution to DT limitations and how much of a difference it might make. Please refer to Garrigues et al. (2022) and relevant references in that study.
We have added the paper in the discussion. Actually, CAMS does use DB retrievals to some extent.

L173: Remove the "tau" symbol after "AOD" and the "lambda" symbol after "wavelength." Those symbols haven't been used anywhere else in the manuscript.
ok

L192-194: It is first mentioned that CCN is averaged daily, but then it is said that the data is computed at 00 UTC.
Yes, CCN is not averaged daily – sorry for that!

Paragraph starting at L191: Instead of "CCN", it seems more proper to use "aerosol number concentration" for the first paragraph of Section 2.2.
changed to "CCN number concentration"

Paragraph starting at L201: To make a clear transition to the new topic, you can add something like this at the beginning of the paragraph: "After deriving the aerosol number concentration of aerosol species from the CAMS reanalysis, CCNs are calculated diagnostically…" (If I understand correctly)
The box model does not read aerosol number concentration, but aerosol mass mixing ratios directly from CAMSRA. So this would not fit here.

L209: Please define the geometric standard deviation in one sentence.
The geometric standard deviation is a dimensionless multiplicative factor of the geometric mean to describe the spread of the distribution. The geometric mean is commonly used for logarithmic distributions and can be expressed as the exponential of the arithmetic mean of logarithms.

L210: I don't think it is accurate to say Equation 2 is derived by dividing density by mass. It is an assumption for the particle size distribution.
We have rephrased this part to avoid misunderstandings.

L234: What do "w" and "a" stand for in the subscript of tension parameter? Are they "water" and "air"?
Originally yes, but we have changed w to s, so $\sigma_{s/a}$ is the surface tension parameter of the solution/air interface to be more conform with previous notations.

L240-242:
- I encourage you to rephrase this part. What I understand from this is that although supersaturation in warm clouds typically ranges between 0.1% and 1.5% (Spracklen et al., 2011), you selected values near the lower limit of s (Pohlker et al., 2023). Therefore, s << 1 is valid. Is that correct?
- Also, how is it justified to select the lower limit of s for warm clouds?

- Additionally, you mentioned in L195 that s is selected from 0.1 to 1% in your dataset. Most of these values do not satisfy (s << 1). An explanation would be helpful.

Sorry for the misunderstanding here. s actually refers to supersaturation in percentage, but we also used s for deviation of the saturation ratio $S = e / e_s$. We now explicitly distinguished throughout the entire manuscript between supersaturation s (in %), as used e.g. for Twomey's power law in the appendix or in the text for figure description, and supersaturation $\Delta S$ as a deviation from the saturation ratio. Example: when saturation is reached then $S = 1$, $\Delta S = 0$ and $s = 0\%$. For a supersaturation of $s = 1\%$, $\Delta S = 0.01$ and S would be 1.01. Thus, you see that we did not just select the lower limit of supersaturations, but actually apply the entire range and that the values satisfy the stated condition.

Equations 7 and 8: It seems that these are based on radius, whereas equations 3-6 are based on diameter. Is adding Equations 7 and 8 necessary? There is a simple relationship between diameter and radius that can be applied at the very end.

Yes, equation 7 is not necessary. In the Literature it often switches back and forth. However, since we only refereed to diameters before, it is not necessary to jump to radii here. So we now tried to be as consistent as possible.

Figure 1: What is the reason for averaging over 2007? Why not use the full climatology? Also, it looks like high-elevation regions show lower CCN. Is this because the CCN is integrated over a column with smaller height for these regions? If so, it would be nice to see maps of surface CCN and address this artefact. You can include these in the supplementary.

The Figure is changed to show the 19-year average. Indeed, high-elevation regions show lower CCN concentrations, but as you can see from Figure C1 (average over lowest 1km above surface) in comparison to Figure 1 (integral over the lowest 10km above surface), this is not due to the vertical integral for which we of course use weighted layer dephts. We also see the same feature when only using the lowest model level using sigma-hybrid pressure (not shown). We elaborate about the reason behind this in the text and refer to other studies.

L270: Perhaps mention that CCN load is the vertical integration of CCN over all levels up to 10 km.
Ok

L273-274: One sentence to describe the reason for this would be great.
OK. Also see Figure C2,b.

L276-277: I am not sure if this can be seen in the vertical profiles. Maybe mentioning the exact latitude and height can help. I don't see it in maps either.
We are referring to this "dip" in the vertical profiles around 5N, mostly noticable between 2 and 4 km height. The effect might be too weak to show on the maps of CCN load with the chosen color scale.

L280-281: Is this due to small CCNs that correspond to larger supersaturation? See Figure 7 in Che et al. (2017).
Yes, the larger the supersaturation, the more CCN get activated + smaller, light-weighted particles can reach higher altitudes by convection and turbulent mixing than larger, heavier particles.

L282: Is this CCN decoupling or just very low CCN concentration over Antarctica? It might be good to briefly define the West Wind Drift.
We have rewritten this sentence. Both poles have CCN concentrations, but in contrast to Antarctica, the Arctic is impacted from pollution advection from lower latitudes.

Figure 2b: When you average over all latitudes for each longitude, it is possible that some high values of N.H. summer cancel low values of S.H. winter. This might lead to less distinction among seasons on this panel.
That could be, but still, we don't see a reason why we should not show this distribution.

L286-287: If you have checked figures for MO and SU values, please mention "figure not shown" after MO and SU.
We added references here to other figures.

L290: It seems to me that there is a seasonal cycle, but overall, the CCN concentration is low over the Pacific. Also, did you mean 140W?
Sorry that was actually East Atlantic, but we deleted this sentence anyway.

L296-297: Can this be confirmed in your data by looking at time series of different components?
We have added the component timeseries to the Appendix, Figure C4, and added some description referring to them in this paragraph. The features described can also be seen in the various components.

L312: The IMO 2020 regulation limits sulfur in ship's fuel. Is that implemented in the CAMS model, and if so, is it detectable in the SU component of CCN?
This should be included in one of the emission inventories . See
https://atmosphere.copernicus.eu/anthropogenic-and-natural-emissions
and more specifically
https://atmosphere.copernicus.eu/sites/default/files/publications/CAMS261_2021SC1_D6.1.2-2022_202306_Docu_v1_APPROVED_Ver1.pdf

However, if one is interested to see if a specific aerosol emission or regulation has an effect on CCN or CCN component, we encourage to analyse the data themselves. This would go beyond the scope of this manuscript.

L315: Do you mean OM? OC has not been mentioned before.
We refer here to Organic carbon from the Quaas et al. (2022) study.

Figure 4: Again, why not show 19-year averages?
We have changed that, see previous comments

L335: Please add "North Hemispheric" before "high latitudes."
OK

L345-346: Please refer to Figure 2a.
Referred to Figure 6a.

L350-351: This sentence doesn't seem correct and can be deleted or rephrased.
Yes, we deleted this sentence as we actually do not analyse anthropogenic contribution which is implicit to the data.

L353 and L371-374: Is there a reference for ARM datasets and their preparation and quality control?
We have added links to the ARM website where the data can be accessed in the caption of Table 2. They also give Quality reports if available and describe the data handling, but there is not a citable reference we are aware of. For the MAGIC campaign, another data product was used and we have

specified this a bit more in the text, giving also an additional reference on the instrument handbook. By doing this, we have noticed that the dataset listed in the table was mixed up with the facility code. This has been corrected.

L378-379: Please add that on average CAMS overestimates CCN for three stations (MAG, PGH, PVC) and this is stronger for PGH with the largest CCN.
ok

Figure 7: Related to previous comment, is there an explanation for CAMS CCN overestimation or does it depend on the sites? Have you seen evidence that CAMS produces underestimated CCN?
We have just started to compare with CCN retrieved from at ACTRIS surface sites (data retrieved from figshare), not publishable at the moment – so this is just a quick preview! CAMS does reasonably well also for these so far 3 stations (HYY, VAV & JFJ), overestimating one station and underestimating two stations. For one station it underestimates CCN heavily (ZEP), however this is outside the assimilation limit of 70N.

[Figure]

[Figure]

Furthermore, we started to compare this dataset to CCN profiles retrieved from CALIOP lidar observations and this also looks like near-surface CCN are lower than those from the active sensor (also, not a publishable result yet!). So, yes, CAMS CCN might underestimate as well. But we need more observations for comparison to get a better understanding of the actual biases which we can then address in future versions of this dataset.

L379: Add "as shown by dashed lines" after "within a factor of 10".
OK

L381: The spread of data seems larger for MAG because it is on the lower end of the graph on a log-log scale. In fact, the spread of data for PGH spans more than 5000 cm-3 (at least for CAMS CCN), whereas the MAG data spread is ~ 2000 cm-3.
Even if the spread in absolute values is higher for the PGH site, in relative terms it is actually a bit smaller than for the MAG site considering the different mean scales of observed CCN. We rephrase this.

L384: Some changes are Figure 7b reveals almost balanced contributions from SU and OM for SGP and PVC, OM dominance for PGH, and SU dominance for GRW and MAG.
Ok, we have changed this.

L387: PGH doesn't seem to be an exception. Perhaps, delete "except for PGH."
In Table 3, the bias factor of median values is 0.93 for SGP, and the bias factor for the interquartile range is 0.89. These values are below 1, meaning an underestimation even though very slighty. All other values are above one, that is an overestimation.

Table 3 Caption:
- Perhaps refer to the text where you define observational measurement uncertainties and select the value of 40%. Is this uncertainty arbitrary? Depending on this value, the CAMS CCN can be within or outside the range of observational uncertainty.
Yes, we think this value is chosen rather arbitrarily in Spracklen et al (2011) where they choose the highest uncertainty value found in other studies to be applicable for their data synthesis, but also commenting that mostly the range is within 10-20%. They further argue, that in any case the uncertainties of the observational data are smaller than the studied effect of aerosols on simulated CCN concentrations. To be on the safe side, we follow their lines and choose a rather high uncertainty range rather than be too low. However, even if the range would be half of that, it would not change the mean bias ratios of the various percentiles.

- Can you add a brief description of how to interpret the 'bias' and NRMSE? For example, a value of 1 for 'bias factor' represents the best agreement.
We added some more explanation in the table caption.

- Perhaps the use of "bias factor" or "bias ratio" is more accurate?
Yes, we changed to bias ratio.

- What does 'Q' in Q50, etc., refer to? If it stands for quartile, it should be Q1, Q2, Q3, and Q4, which correspond to P25, P50, P75, and P100 (P refers to percentile).
OK, we have changed Q for P as we refer to percentiles here.

L390-395: This highlights a possible source of bias in reanalysis and GCMs.
- This bias might not be evident from total AOD (Figure 20, first row, third column in Inness et al., 2019), but the substantial positive difference in OM AOD is clearly evident in the third row, third column.
- This effect is more pronounced over China but can also be observed to a lesser extent over Europe and the Eastern USA (Figure 20 in Inness et al., 2019).
Yes indeed! We have added some more lines concerning this feature.

- It would greatly reinforce your findings to include brief comparisons with other studies that compare models and observations for these specific regions (assuming such studies exist).
We have added a reference comparing the PGH ARM CCN values to another version of modeled CCN using MACCRA data. This emphasizes that something is off with the CAMS data.

Table 4: Including correlation analysis is helpful. A few questions:
- What is the number of data points for each site? I noticed that the total value in Table 4 is approximately half that in Table 3. Does this suggest that some sites (e.g., MAG) have a very low number of data points? Adding the number of data points as a new column to Table 4 would be informative.
Ok, we have added more information to this Table.

- Additionally, could you consider showing the statistical significance (or p-value) for each correlation? There are look-up tables that provide p-values based on correlation and the number of data points (or degree of freedom). Incorporating this information would enhance the analysis, as the correlation value can be influenced by the number of data points.
Yes, we included that now.

L400: Perhaps rephrase as 'The correlation coefficient for all data points increases from 0.37 to 0.71...
ok

L401: Considering the previous point, it might not be appropriate to directly compare changes in correlation among different sites, as they have varying numbers of data points. Additionally, the change from 0.07 to 0.18 doesn't necessarily represent a substantial improvement due to the overall weak correlation. Nevertheless, the following discussion based on previous studies is insightful and worth retaining.
We have rephrased the paragraph analysing this table.

L415-418: You can omit the first two sentences and place the last one at the end of the first sentence in L401.
ok

L420: You can provide a more accurate statement by saying, 'Since the reanalysis aerosol mass mixing ratios are generated through the assimilation of satellite AOD…
ok

L443: Change to something like 'The contribution of black carbon to CCN is significantly smaller than that of sulfate and organic matter…
ok

L444: You may consider deleting this line or providing a more specific and detailed explanation.
It is deleted.

L472: The latter part of this sentence is difficult to understand.
We have rephrased this.

Technical Corrections:
Ensure that acronyms and abbreviations are defined in their first occurrence in the manuscript. Some examples:
L8: Replace "RA" with "(CAMSRA)". Also, "EAC4" is not clear to me.
Eac4 stands for "earth atmospheric composition experiment 4", it is a version number of ECMWF reanalysis. We actually don't need it in the text. Now it only occurs in the links and references.

L24: Define AERONET
ok

L45: Define ECHAM-HAM
ok

L50: Define POLDER
ok

L59: Define MAC
ok

L67: Change "Caliop" to "the Cloud-Aerosol Lidar with Orthogonal Polarization (CALIOP)"

ok

L85: GCM should be defined here.
Replaced by "climate models" to generalize

L104-105: Define LMD-Z, GEMS, MACC.
ok

L143-144: Define MEGAN2.1 and MERRA-2
ok

L201: Define HadGEM3-UKCA
ok

Appendix B: Define "CN" and "NC".
Theres was a mixup between NC and CN which we have corrected.

Grammar and Typos:
L1-2: Consider rephrasing to correct the grammar, such as adding "and" before "a process."
ok

L56: Change "related with" to "related to."
ok

L91: Change "Deviations to" to "Deviations from."
ok

L162: Change "on board of the Aqua" to "on board the Aqua."
ok

L262: Change "radii is" to "radii are."
ok, now diameters

L278-279: Change "stay" and "decrease" to "stays" and "decreases."
ok

L296: Change "where" to "were."
ok

L299: Change "eruption on Iceland" to "eruption in Iceland."
ok

Figure 3 caption: Change "latitudional" to "latitudinal" and "timeseries" to "time-series."
ok

L331: Change "wind speed are" to "wind speeds are."
ok

Figure 8 caption: Remove the extra "the" in "the the ARM."
ok

L519: Change "is" to "are" in "A demonstration and a possible extension of the formula is given."
ok

L527: Change "Thus study" to "This study."
ok

L535: Change "office of science" to "Office of Science."
ok

Use hyphens when two words function together as an adjective before a noun. Some examples:
L5: Change "uncertainty reduced estimates" to "uncertainty-reduced estimates".
ok

L14: Change "ground based in-situ measurements" to "ground-based in-situ measurements".
ok

L38: Change "satellite retrieved AOD" to "satellite-retrieved AOD".
ok

L58: Change "CCN related aerosol" to "CCN-related aerosol".
ok

L81 and similar occurrences: Change "CCN relevant" to "CCN-relevant" throughout the manuscript.
ok

L134: Change "latitude dependent e-folding time scale" to "latitude-dependent e-folding time scale".
ok

L191 and 419: Change "19 year long global CCN" to "19-year-long global CCN"
ok

L353 and similar occurrences: Change "CAMS derived" to "CAMS-derived" and "quality assured" to "quality-assured" throughout the manuscript.
ok

L359: Change "pyramid like" to "pyramid-like".
ok

L425: Change "cloud free" to "cloud-free".
ok

L493: Change "quality checked" to "quality-checked".
ok

Be sure to be consistent about past tense and present tense throughout the manuscript.
ok

Example:
L375: Perhaps, change "ensured" to "ensure" in "We use daily means of quality-checked CCN measurements and ensured that"
ok

Ensure consistent spelling throughout the manuscript. Decide between British and American English spellings and stick to one style.
ok

Examples:
L125: Change "aging" to "ageing".
ok

L180: Change "parameterized" to "parameterised".
ok

L280 and similar occurrences: Change "analyzed" to "analysed".
ok

References:

Che, H. C., Zhang, X. Y., Zhang, L., Wang, Y. Q., Zhang, Y. M., Shen, X. J., ... & Zhong, J. T. (2017). Prediction of size-resolved number concentration of cloud condensation nuclei and long-term measurements of their activation characteristics. Scientific reports, 7(1), 5819. https://doi.org/10.1038/s41598-017-05998-3

Che, H., Stier, P., Watson-Parris, D., Gordon, H., & Deaconu, L. (2022). Source attribution of cloud condensation nuclei and their impact on stratocumulus clouds and radiation in the south-eastern Atlantic. Atmospheric Chemistry and Physics, 22(16), 10789-10807. https://doi.org/10.5194/acp-22-10789-2022

Garrigues, S., Chimot, J., Ades, M., Inness, A., Flemming, J., Kipling, Z., ... & Agusti-Panareda, A. (2022). Monitoring multiple satellite aerosol optical depth (AOD) products within the Copernicus Atmosphere Monitoring Service (CAMS) data assimilation system. Atmospheric Chemistry and Physics, 22(22), 14657-14692. https://doi.org/10.5194/acp-22-14657-2022

Haarig, M., Walser, A., Ansmann, A., Dollner, M., Althausen, D., Sauer, D., Farrell, D., and Weinzierl, B. (2019). Profiles of cloud condensation nuclei, dust mass concentration, and ice-nucleating-particle-relevant aerosol properties in the saharan air layer over barbados from polarization lidar and airborne in situ measurements. Atmospheric Chemistry and Physics, 19(22):13773–13788. https://doi.org/10.5194/acp-19-13773-2019

Inness, , Ades, M., Agustí-Panareda, A., Barré, J., Benedictow, A., Blechschmidt, … & Suttie, M. (2019). The CAMS reanalysis of atmospheric composition, Atmos. Chem. Phys., 19, 3515–3556, https://doi.org/10.5194/acp-19-3515-2019 .

Kapustin, V. N., Clarke, A. D., Shinozuka, Y., Howell, S., Brekhovskikh, V., Nakajima, T., & Higurashi, A. (2006). On the determination of a cloud condensation nuclei from satellite: Challenges and possibilities. Journal of Geophysical Research: Atmospheres, 111(D4). https://doi.org/10.1029/2004JD005527

Shinozuka, Y., Clarke, A. D., Nenes, A., Jefferson, A., Wood, R., McNaughton, C. S., ... & Yoon, Y. J. (2015). The relationship between cloud condensation nuclei (CCN) concentration and light extinction of dried particles: indications of underlying aerosol processes and implications for satellite-based CCN estimates. Atmospheric Chemistry and Physics, 15(13), 7585-7604. https://doi.org/10.5194/acp-15-7585-2015

Weinzierl, B., Ansmann, A., Prospero, J. M., Althausen, D., Benker,, Chouza, F., … & Walser, A. (2017). The saharan aerosol long-range transport and aerosol–cloud-interaction experiment: Overview and selected highlights. Bulletin of the American Meteorological Society, 98(7):1427 – 1451. https://doi.org/10.1175/BAMS-D-15-00142.1

---

## Author Response (AR2)

Changes for final publications:

1. authors:
Mahnoosh Haghighatnasab's new affiliation was added

2. dataset:
The data doi was added to the abstract (line 18) and in the text (line 223).
Now the dataset is listed as required in abstract, text, data availability section and reference list.

3. color schemes:
As requested we have adjusted all color schemes to be more color blind friendly by using
https://www.color-blindness.com/coblis-color-blindness-simulator/. The content or description of the plots did not change!

4. Figure 7(b):
Instead of using 3 types of sea salt (SS) being displayed, we chose to show sea salt as a single component which naturally consists of the three types. This is easier as 1) sea salt does actually not show any noticeable contribution to this bar chart after all and it would be exaggerated to separate various types, and 2) this way the colors are consistent to the other plots where the four CCN species (SU, BC, OM & SS) are distinguished.